# Optimal Inference in Contextual Stochastic Block Models

**O. Duranthon and L. Zdeborová**
**Statistical Physics of Computation laboratory,**
**École polytechnique fédérale de Lausanne (EPFL), Switzerland**
`firstname.lastname@epfl.ch`

**Reviewed on OpenReview:** `https://openreview.net/forum?id=Pe6hldOUkw`

## Abstract

The contextual stochastic block model (CSBM) was proposed for unsupervised community detection on attributed graphs where both the graph and the high-dimensional node information correlate with node labels. In the context of machine learning on graphs, the CSBM has been widely used as a synthetic dataset for evaluating the performance of graph-neural networks (GNNs) for semi-supervised node classification. We consider a probabilistic Bayes-optimal formulation of the inference problem and we derive a belief-propagation-based algorithm for the semi-supervised CSBM; we conjecture it is optimal in the considered setting and we provide its implementation. We show that there can be a considerable gap between the accuracy reached by this algorithm and the performance of the GNN architectures proposed in the literature. This suggests that the CSBM, along with the comparison to the performance of the optimal algorithm, readily accessible via our implementation, can be instrumental in the development of more performant GNN architectures.

## 1 Introduction

In this paper we are interested in the inference of a latent community structure given the observation of a sparse graph along with high-dimensional node covariates, correlated with the same latent communities. With the same interest, the authors of Yan & Sarkar (2021); Deshpande et al. (2018) introduced the contextual stochastic block model (CSBM) as an extension of the well-known and broadly studied stochastic block model (SBM) for community detection. The CSBM accounts for the presence of node covariates; it models them as a high-dimensional Gaussian mixture where cluster labels coincide with the community labels and where the centroids are latent variables. Along the lines of theoretical results established in the past decade for the SBM, see e.g. the review Abbé (2017) and references therein, authors of Deshpande et al. (2018) and later Lu & Sen (2020) study the detectability threshold in this model.

Our motivation to study the CSBM is due to the interest this model has recently received in the community developing and analyzing graph neural networks (GNNs). Indeed, this model provides an idealized synthetic dataset on which graph neural networks can be conveniently evaluated and benchmarked. It has been used, for instance, in Baranwal et al. (2021); Kimon et al. (2022); Javaloy et al. (2023); Shi et al. (2023) to establish and test theoretical results on graph convolutional networks or graph-attention neural networks. In Wu et al. (2023) the CSBM was used to study over-smoothing of GNNs and in Wei et al. (2022) to study the role of non-linearity. As a synthetic dataset the CSBM has also been utilized in Cong et al. (2021) for supporting theoretical results on depth in graph convolutional networks and in Chien et al. (2021); Fu et al. (2021); Lei et al. (2022) for evaluating new GNN architectures. Some of the above works study the CSBM in the unsupervised case; however, more often they study it in the semi-supervised case where on top of the network and covariates we observe the membership of a fraction of the nodes.

While many of the above-cited works use the CSBM as a benchmark and evaluate GNNs on it, they do not compare to the optimal performance that is tractably achievable in the CSBM. A similar situation happened in the past for the stochastic block model. Many works were published proposing novel community detection algorithms and evaluating them against each other, see e.g. the review Fortunato (2010) and references

there-in. The work of Decelle et al. (2011) changed that situation by conjecturing that a specific variant of the belief propagation algorithm provides the optimal performance achievable tractably in the large size limit. A line of work followed where new algorithms, including early GNNs (Chen et al., 2020), were designed to approach or match this predicted theoretical limit.

The goal of the present work is to provide a belief-propagation-based algorithm, which we call AMP–BP, for the semi-supervised CSBM. It is able to deal with noisy-labels; it can estimate the parameters of the model and we propose a version of it for multiple unbalanced communities. We conjecture AMP–BP has optimal performance among tractable algorithms in the limit of large sizes. We provide a simple-to-use implementation of the algorithm (attached in the Supplementary Material) so that researchers in GNNs who use CSBM as a benchmark can readily compare to this baseline and get an idea of how far from optimality their methods are. We also provide a numerical section illustrating this, where we compare the optimal inference in CSBM with the performance of some of state-of-the-art GNNs. We conclude that indeed there is still a considerable gap between the two; and we hope the existence of this gap will inspire follow-up work in GNNs aiming to close it.

**Related work**    Previous works deal with optimal inference in CSBM. In a setting with low-dimensional features there is Braun et al. (2022) that is unsupervised for multi-community; it does not require hyper-parameter tuning but it focuses on exact recovery when the graph signal-to-noise ratio grows with number of nodes. Abbé et al. (2022) proposes an optimal spectral algorithm for unsupervised binary CSBM in the case of growing degrees. There is also Baranwal et al. (2023) that determines a localy-optimal semi-supervised classifier and shows it can interpreted as a GNN. In the same high-dimensional setting as ours, there are Lu & Sen (2020) and the belief-propagation-based algorithm Deshpande et al. (2018); they are unsupervised and need the right parameters of the model; we discuss them more in detail further. Works on optimality in related models of SBMs with features include Duranthon & Zdeborová (2023), where the group memberships are a generic function of the features.

We consider the CSBM in a semi-supervised high-dimensional sparse setting. Semi-supervision is necessary because we want to compare to empirical risk minimizers such that GNNs; they require a fraction of train labels to be trained. In the regime where the features are high-dimensional, many quantities of interest converge to deterministic values and allow a precise analysis, as opposed to low-dimensional features. We choose a setting with a sparse graph because this is closer to what in practice researchers in GNNs use. The setting where the graph is dense is theoretically easier to deal with (e.g. it has been analyzed in Deshpande et al. (2018) and Shi et al. (2023)) and we provide the corresponding optimal algorithm in appendix.

## 2 Setup

### 2.1 Contextual stochastic block model (CSBM)

We consider a set $V$ of $|V| = N$ nodes and a graph $G(V, E)$ on those nodes. Each of the nodes belongs to one of two groups: $u_i \in \{-1, +1\}$ for $i = 1, \ldots, N$. We draw their memberships independently, and we consider two balanced groups: $u_i$ is Rademacher. We make this choice following previous papers that used CSBM to study graph neural networks. We note, however, that multiple communities or arbitrary sizes can be readily considered, as done for the SBM in Decelle et al. (2011) and for the high-dimensional Gaussian mixture e.g. in Lesieur et al. (2017). In appendix D we define the unbalanced mulity-community setup and provide the corresponding AMP–BP algorithm.

The graph is generated according to a stochastic block model (SBM):

$$P(A_{ij} = 1 | u_i, u_j) = \left\{ \begin{array}{ll} c_i/N & \text{if} \quad u_i = u_j, \\ c_o/N & \text{if} \quad u_i \neq u_j, \end{array} \right. \tag{1}$$

and $A_{ij} = 0$ otherwise. $c_i \in \mathbb{R}$ and $c_o \in \mathbb{R}$ are the two affinity coefficients common to the SBM. We stack them in the matrix $C = \left( \begin{smallmatrix} c_i & c_o \\ c_o & c_i \end{smallmatrix} \right)$.

We also consider feature/attribute/covariate $B \in \mathbb{R}^{P \times N}$ of dimension $P$ on the $N$ nodes. They are generated according to a high-dimensional Gaussian mixture model:

$$B_{\beta i} = \sqrt{\frac{\mu}{N}} v_\beta u_i + Z_{\beta i} \tag{2}$$

for $\beta = 1, \ldots, P$, where $v_\beta \sim \mathcal{N}(0,1)$ determines the randomly drawn centroids, $Z_{\beta i}$ is standard Gaussian noise and $\mu \in \mathbb{R}$ is the strength of the signal. We precise that there are two centroids: the features of the nodes in the $+1$ group are centered at $\sqrt{\frac{\mu}{N}} v$ while the features of the $-1$ group are centered at $-\sqrt{\frac{\mu}{N}} v$. The edges $A$ and the feature matrix $B$ are observed. We aim to retrieve the groups $u_i$.

We work in the sparse limit of the SBM: the average degree of the graph $A$ is $d = \mathcal{O}(1)$. We parameterize the SBM via the signal-to-noise ratio $\lambda$:

$$c_i = d + \lambda \sqrt{d} \quad ; \quad c_o = d - \lambda \sqrt{d} \,. \tag{3}$$

We further work in the high-dimensional limit of the CSBM. We take both $N$ and $P$ going to infinity with $\alpha = N/P = \mathcal{O}(1)$ and $\mu = \mathcal{O}(1)$.

We define $\Xi$ as the set of revealed training nodes, that are observed. We set $\rho = |\Xi|/N$; $\rho = 0$ for unsupervised learning. We assume $\Xi$ is drawn independently with respect to the group memberships. We define $P_{U,i}$ an additional node-dependant prior. It is used to inject information about the memberships of the observed nodes:

$$P_{U,i}(s) = \left\{ \begin{array}{ll} \delta_{s,u_i} & \text{if} \quad i \in \Xi, \\ 1/2 & \text{if} \quad i \notin \Xi. \end{array} \right. \tag{4}$$

## 2.2 Bayes-optimal estimation

We use a Bayesian framework to infer optimally the group membership $u$ from the observations $A, B, \Xi$. The posterior distribution over the nodes $u = (u_i)_i$ is

$$P(u|A, B, \Xi) = \frac{1}{Z(A, B, \Xi)} P(A|u, B, \Xi) P(B|u, \Xi) P(u|\Xi) \tag{5}$$

$$= \frac{\prod_i P_{U,i}(u_i)}{Z(A, B, \Xi)} \prod_{i<j} P(A_{ij}|u_i, u_j) \int \prod_\beta [\mathrm{d}v_\beta P_V(v_\beta)] \prod_{\beta,i} \frac{1}{\sqrt{2\pi}} e^{-\frac{1}{2}(B_{\beta i} - \sqrt{\frac{\mu}{N}} v_\beta u_i)^2} \,, \tag{6}$$

where $Z(A, B, \Xi)$ is the normalization constant and $P_V = \mathcal{N}(0,1)$ is the prior distribution on $v$. In eq. (6) we marginalize over the latent variable $v = (v_\beta)_\beta$. However, since the estimation of the latent variable $v$ is crucial to infer $u$, it will be instrumental to consider the posterior as a joint probability of the unobserved nodes and the latent variable:

$$P(u, v|A, B, \Xi) = \frac{\prod_i P_{U,i}(u_i) \prod_\beta P_V(v_\beta)}{Z(A, B, \Xi)} \prod_{i<j} P(A_{ij}|u_i, u_j) \prod_{\beta,i} \frac{1}{\sqrt{2\pi}} e^{-\frac{1}{2}(B_{\beta i} - \sqrt{\frac{\mu}{N}} v_\beta u_i)^2} \,, \tag{7}$$

where $Z(A, B, \Xi)$ is the Bayesian evidence. We define the free entropy of the problem as its logarithm:

$$\phi(A, B, \Xi) = \frac{1}{N} \log Z(A, B, \Xi) \,. \tag{8}$$

We seek an estimator $\hat{u}$ that maximizes the mean overlap MO with the ground truth. The Bayes-optimal estimator $\hat{u}$ that maximizes it is given by

$$\mathrm{MO}(\hat{u}) = \sum_u P(u|A, B, \Xi) \frac{1}{N} \sum_{i=1}^N \delta_{\hat{u}_i, u_i} \quad ; \quad \hat{u}_i^{\mathrm{MMO}} = \operatorname*{argmax}_{t=\pm 1} p_i(t) \,, \tag{9}$$

where $p_i$ is the marginal posterior probability of node $i$ i.e. $p_i(t) = \sum_{u,u_i=t} P(u|A,B,\Xi)$. To estimate the latent variable $v$, we consider minimizing the mean squared error MSE via the MMSE estimator

$$\text{MSE}(\hat{v}) = \int dv \sum_u P(u,v|A,B,\Xi)\frac{1}{P}\sum_{\beta=1}^{P}(\hat{v}_\beta - v_\beta)^2 \quad ; \quad \hat{v}_\beta^{\text{MMSE}} = \int \text{d}v \sum_u P(u,v|A,B,\Xi)v_\beta \,, \qquad (10)$$

i.e. $\hat{v}^{\text{MMSE}}$ is the mean of the posterior distribution. Using the ground truth values $u_i$ of the communities and $v_\beta$ of the latent variables, the maximal mean overlap MMO and the minimal mean squared error MMSE are then computed as

$$\text{MMO} = \frac{1}{N}\sum_i \delta_{\hat{u}_i^{\text{MMO}},u_i} \quad ; \quad \text{MMSE} = \frac{1}{P}\sum_\beta (\hat{v}_\beta^{\text{MMSE}} - v_\beta)^2 \,. \qquad (11)$$

In practice, we measure the following test overlap between the estimates $\hat{u}_i$ and the ground truth variables $u_i$:

$$q_U = \frac{\hat{q}_U - 1/2}{1 - 1/2} \quad ; \quad \hat{q}_U = \frac{1}{(1-\rho)N}\max\left(\sum_{i\notin\Xi}\delta_{\hat{u}_i,u_i},\sum_{i\notin\Xi}\delta_{\hat{u}_i,-u_i}\right), \qquad (12)$$

where we rescale $\hat{q}_U$ to obtain an overlap between 0 (random guess) and 1 (perfect recovery) and take into account the invariance by permutation of the two groups in the unsupervised case $\rho = 0$.

In general, the Bayes-optimal estimation requires the evaluation of the averages over the posterior that is in general exponentially costly in $N$ and $P$. In the next section, we derive the AMP–BP algorithm. We argue that the estimators $\hat{u}$ and $\hat{v}$ it computes converge to the MMO and MMSE estimators with a vanishing error: $P(\frac{1}{N}\sum_i \delta_{\hat{u}_i^{\text{MMO}},\hat{u}_i} < 1 - \epsilon) \underset{N\to\infty}{\to} 0$ and $P(\frac{1}{P}\sum_\beta (\hat{v}_\beta^{\text{MMSE}} - \hat{v}_\beta)^2 > \epsilon) \underset{N\to\infty}{\to} 0$ for any $\epsilon > 0$ with $N/P = \alpha = \mathcal{O}(1)$ and all other parameters being of $\mathcal{O}(1)$.

**Detectability threshold and the effective signal-to-noise ratio**  Previous works on the inference in the CSBM (Deshpande et al., 2018; Lu & Sen, 2020) established a detectability threshold in the unsupervised case, $\rho = 0$, to be

$$\lambda^2 + \frac{\mu^2}{\alpha} = 1\,. \qquad (13)$$

meaning that for a signal-to-noise ratio smaller than this, it is information-theoretically impossible to obtain any correlation with the ground truth communities. On the other hand, for snr larger than this, the works Deshpande et al. (2018); Lu & Sen (2020) demonstrate algorithms that are able to obtain a positive correlation with the ground truth communities.

This detectability threshold also intuitively quantifies the interplay between the parameters, the graph-related snr $\lambda$ and the covariates-related snr $\mu^2/\alpha$. Small $\mu^2/\alpha$ generates a benchmark where the graph structure carries most of the information; while small $\lambda$ generates a benchmark where the information from the covariates dominates; and if we want both to be comparable, we consider both comparable. The combination from eq. (13) plays the role of an overall effective snr and thus allows tuning the benchmarks between regions where getting good performance is challenging or easy.

The threshold (13) reduces to the one well known in the pure SBM (Decelle et al., 2011) when $\mu = 0$ and to the one well known in the unsupervised high-dimensional Gaussian mixture (Lesieur et al., 2016) when $\lambda = 0$.

## 3  The AMP–BP Algorithm

We derive the AMP–BP algorithm starting from the factor graph representations of the posterior (7):

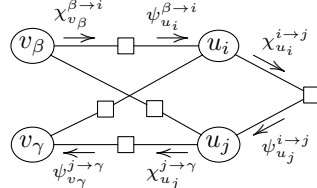

The factor graph has two kinds of variable nodes, one kind for $v$ and the other one for $u$. The factors are of two types, those including information about the covariates $B$ that form a fully connected bipartite graph between all the components of $u$ and $v$, and those corresponding to the adjacency matrix $A$ that form a fully connected graph between the components of $u$.

We write the belief-propagation (BP) algorithm for this graphical model (Yedidia et al., 2003; Mézard & Montanari, 2009). It iteratively updates the so-called messages $\chi$s and $\psi$s. These different messages can be interpreted as probability distributions on the variables $u_i$ and $v_\beta$ conditioned on the absence of the target node in the graphical model. The iterative equations read (Yedidia et al., 2003; Mézard & Montanari, 2009)

$$\chi_{v_\beta}^{\beta \to i} \propto P_V(v_\beta) \prod_{j \neq i} \psi_{v_\beta}^{j \to \beta} \ , \qquad\qquad \psi_{u_i}^{\beta \to i} \propto \int \mathrm{d}v_\beta \, \chi_{v_\beta}^{\beta \to i} e^{-(B_{\beta i} - w_{\beta i})^2/2} \ , \qquad (14)$$

$$\chi_{u_i}^{i \to j} \propto P_{U,i}(u_i) \prod_{\beta} \psi_{u_i}^{\beta \to i} \prod_{k \neq i,j} \psi_{u_i}^{k \to i} \ , \qquad\qquad \psi_{u_j}^{i \to j} \propto \sum_{u_i} \chi_{u_i}^{i \to j} P(A_{ij} | u_i, u_j) \ , \qquad (15)$$

$$\chi_{u_j}^{j \to \gamma} \propto P_{U,j}(u_j) \prod_{\beta \neq \gamma} \psi_{u_j}^{\gamma \to j} \prod_{k \neq j} \psi_{u_j}^{k \to j} \ , \qquad\qquad \psi_{v_\gamma}^{j \to \gamma} \propto \sum_{u_j} \chi_{u_j}^{j \to \gamma} e^{-(B_{\gamma j} - w_{\gamma j})^2/2} \ , \qquad (16)$$

where $w_{\beta i} = \sqrt{\frac{\mu}{N}} v_\beta u_i$ for all $i$ and $\beta$ and where the proportionality sign $\propto$ means up to the normalization factor that ensures the message sums to one over its lower index.

We conjecture that the BP algorithm is asymptotically exact for CSBM. BP is exact on graphical models that are trees, which the one of CSBM is clearly not. The graphical model of CSBM, however, falls into the category of graphical model for which the BP algorithm for Bayes-optimal inference is conjectured to provide asymptotically optimal performance in the sense that, in the absence of first-order phase transitions, the algorithm iterated from random initialization reaches a fixed point whose marginals are equal to the true marginals of the posterior in the leading order in $N$.

This conjecture is supported by previous literature. The posterior (7) of the CSBM is composed of two parts that are independent of each other conditionally on the variables $u$, the SBM part depending on $A$, and the Gaussian mixture part depending on $B$. Previous literature proved the asymptotic optimality of the corresponding AMP for the Gaussian mixture part in Dia et al. (2016). As to the SBM part, the asymptotic optimality of BP (Decelle et al., 2011) was proven for the binary semi-supervised SBM in Yu & Polyanskiy (2022). Because of the conditional independence, the optimality is expected to be preserved when we concatenate the two parts into the CSBM. For the sparse standard SBM in the unsupervised case the conjecture remains mathematically open.

The above BP equations can be simplified in the leading order in $N$ to obtain the AMP–BP algorithm. The details of this derivation are given in appendix A. This is done by expanding in $w$ in part accounting for the high-dimensional Gaussian mixture side of the graphical model. This is standard in the derivation of the AMP algorithm, see e.g. Lesieur et al. (2017). On the SBM side the contributions of the non-edges are concatenated into an effective field, just as it is done for the BP on the standard SBM in Decelle et al. (2011). The AMP–BP algorithm then reads as in Algorithm 1. A version of the algorithm for an unbalanced mulity-community setup is given in section D in the appendix.

To give some intuitions we explain what are the variables AMP–BP employs. The variable $\hat{v}_\beta$ is an estimation of the posterior mean of $v_\beta$, whereas $\sigma_V$ of its variance. The variable $\hat{u}_i$ is an estimation of the posterior mean of $u_i$, $\sigma_U$ of its variance. Next $B_U^\beta$ is a proxy for estimating the mean of $v_\beta$ in the absence of the Gaussian mixture part, $A_U$ for its variance; $B_V^i$ is a proxy for estimating the mean of $u_i$ in absence of the SBM part, $A_V$ for its variance. Further $h_u$ can be interpreted as an external field to enforce the nodes not to be in the same group; $\chi_+^{i \to j}$ is a marginal distribution on $u_i$ (these variables are the messages of a sum-product message-passing algorithm); and $\chi_+^i$ is the marginal probability that node $i$ is $+1$, that we are interested in.

The AMP–BP algorithm can be implemented very efficiently: it takes $\mathcal{O}(NP)$ in time and memory, which is the minimum to read the input matrix $B$. Empirically, the number of steps to converge does not depend on $N$ and is of order ten, as shown on Fig. 6 in appendix F. We provide a fast implementation of AMP–BP

**Input:** features $B_{\beta i} \in \mathbb{R}^{P \times N}$, observed graph $G$, affinity matrix $C$, prior information $P_{U,i}$.

**Initialization:** for $(ij) \in G$, $\chi_+^{i \to j,(0)} = P_{U,i}(+) + \epsilon^{i \to j}$, $\hat{u}_i^{(0)} = 2P_{U,i}(+) - 1 + \epsilon^i$, $\hat{v}_\beta^{(0)} = \epsilon^\beta$, $t = 0$, where $\epsilon^{i \to j}$, $\epsilon^i$ and $\epsilon^\beta$ are small centered random variables in $\mathbb{R}$.

**Repeat until convergence:**

$$\sigma_U^i = 1 - \hat{u}_i^{(t),2}$$

AMP estimation of $\hat{v}$

$$A_U = \frac{\mu}{N} \sum_i \hat{u}_i^{(t),2}$$

$$B_U^\beta = \sqrt{\frac{\mu}{N}} \sum_i B_{\beta i} \hat{u}_i^{(t)} - \frac{\mu}{N} \sum_i \sigma_U^i \hat{v}_\beta^{(t)}$$

$$\hat{v}_\beta^{(t+1)} \leftarrow B_U^\beta / (1 + A_U)$$

$$\sigma_V = 1/(1 + A_U)$$

AMP estimation of $\hat{u}$

$$B_V^i = \sqrt{\frac{\mu}{N}} \sum_\beta B_{\beta i} \hat{v}_\beta^{(t+1)} - \frac{\mu}{\alpha} \sigma_V \hat{u}_i^{(t)}$$

Estimation of the field $h$

$$h_u = \frac{1}{N} \sum_i \sum_{t=\pm 1} C_{u,t} \frac{1 + t\hat{u}_i^{(t)}}{2}$$

$$\tilde{h}_u^i = -h_u + \log P_{U,i}(u) + u B_V^i$$

$C_{u,t}$ being the affinity between groups $u$ and $t$.
BP update of the messages $\chi^{i \to j}$ for $(ij) \in G$ and of marginals $\chi^i$

$$\chi_+^{i \to j,(t+1)} \leftarrow \sigma \left( \tilde{h}_+^i - \tilde{h}_-^i + \sum_{k \in \partial i \setminus j} \log \left( \frac{c_o + 2\lambda \sqrt{d} \chi_+^{k \to i,(t)}}{c_i - 2\lambda \sqrt{d} \chi_+^{k \to i,(t)}} \right) \right)$$

$$\chi_+^i = \sigma \left( \tilde{h}_+^i - \tilde{h}_-^i + \sum_{k \in \partial i} \log \left( \frac{c_o + 2\lambda \sqrt{d} \chi_+^{k \to i,(t)}}{c_i - 2\lambda \sqrt{d} \chi_+^{k \to i,(t)}} \right) \right)$$

where $\sigma(x) = 1/(1 + e^{-x})$ is the sigmoid and $\partial i$ are the nodes connected to $i$.
BP estimation of $\hat{u}$

$$\hat{u}_i^{(t+1)} \leftarrow 2\chi_+^i - 1$$

Update time $t \leftarrow t + 1$.
**Output:** estimated groups $\mathrm{sign}(\hat{u}_i)$.

**Algorithm 1:** *The AMP–BP algorithm.*

written in Python in the supplementary material and in our repository.[1] The algorithm can be implemented in terms of fast vectorized operations as to the AMP part; and, as to the BP part, vectorization is possible thanks to an encoding of the sparse graph in an $\mathcal{O}(Nd) \times \mathcal{O}(d)$ array with a padding node. Computationally, running the code for a single experiment, $N = 3 \times 10^4$, $\alpha = 1$ and $d = 5$ takes around one minute on one CPU core.

We cross-check the validity of the derived AMP-BP algorithm by independent Monte-Carlo simulations. We sample the posterior distribution (7) with the Metropolis algorithm; the estimates for the communities are then $\mathrm{sign}(\sum_t u_i^t)$, $u^t$ being the different samples. As shown on Fig. 7 in appendix F, the agreement with AMP–BP is very good except close to small $q_U$, i.e. close to the phase transition, where the Markov chain seems to take time to converge.

**Related work on message passing algorithms in CSBM** The AMP–BP algorithm was stated for the unsupervised CSBM in section 6 of Deshpande et al. (2018) where it was numerically verified that it indeed presents the information-theoretic threshold (13). In that paper, little attention was given to the performance of this algorithm besides checking its detectability threshold. In particular, the authors did not comment on the asymptotic optimality of the accuracy achieved by this algorithm. Rather, they linearized it and studied the detectability threshold of this simplified linearized version that is amenable to analysis

---
[1]gitlab.epfl.ch/spoc-idephics/csbm

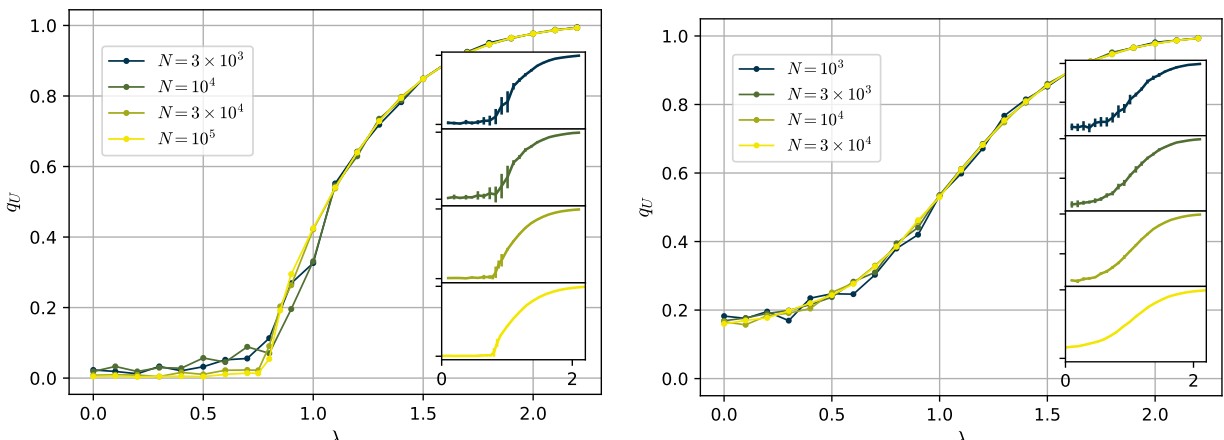

Figure 1: *Convergence to the high-dimensional limit.* Overlap $q_U$ of the fixed point of AMP–BP vs the snr $\lambda$ for several system sizes $N$. *Left:* unsupervised case, $\rho = 0$. *Right:* semi-supervised, $\rho = 0.1$. The other parameters are $\alpha = 10$, $\mu^2 = 4$, $d = 5$. We run ten experiments per point.

via random matrix theory. This threshold matches the information-theoretical detectability threshold that was later established in Lu & Sen (2020). The linearized version of the AMP–BP algorithm is a spectral algorithm; it has sub-optimal accuracy, as we will illustrate below in section 3. We also note that the work Lu & Sen (2020) considered another algorithm based on self-avoiding walks. It reaches the threshold but it is not optimal in terms the overlap in the detectable phase or in the semi-supervised case, nor in terms of efficiency since it quasi-polynomial. Authors of Deshpande et al. (2018); Lu & Sen (2020) have not considered the semi-supervised case of CSBM, whereas that is the case that has been mostly used as a benchmark in the more recent GNN literature.

**Bayes-optimal performance** We run AMP–BP and show the performance it achieves. Since the conjecture of optimality of AMP–BP applies to the considered high-dimensional limit, we first check how fast the performance converges to this limit. In Fig. 1, we report the achieved overlap when increasing the size $N$ to $+\infty$ while keeping the other stated parameters fixed. We conclude that taking $N = 3 \times 10^4$ is already close to the limit; finite-size effects are relatively small.

Fig. 2 shows the performance for several different values of the ratio $\alpha = N/P$ between the size of the graph $N$ and the dimensionality of the covariates $P$. Its left panel shows the transition from a non-informative fixed point $q_U = 0$ to an informative fixed point $q_U > 0$, that becomes sharp in the limit of large sizes. It occurs in the unsupervised regime $\rho = 0$ for $\alpha$ large enough. The transition is located at the critical threshold $\lambda_c$ given by eq. (13). This threshold is shared by AMP–BP and the spectral algorithm of Deshpande et al. (2018) in the unsupervised case. The transition is of 2nd order, meaning the overlaps vary continuously with respect to $\lambda$. As expected from statistical physics, the finite size effects are stronger close to the threshold; this means that the variability from one experiment to another one is larger when close to $\lambda_c$.

The limit $\alpha \to +\infty$, in our notation, leads back to the standard SBM, and the phase transition is at $\lambda = 1$ in that limit. Taking $\alpha \leq \mu^2$ or adding supervision $\rho > 0$ (Fig. 2 right) makes the 2nd order transition in the optimal performance disappear.

The spectral algorithm given by Deshpande et al. (2018) is sub-optimal. In the unsupervised case, it is a linear approximation of AMP–BP, and the performances of the two are relatively close. In the semi-supervised case, a significant gap appears because the spectral algorithm does not naturally use the additional information given by the revealed labels; it performs as if $\rho = 0$.

**Dense limit** We can consider the dense limit of the CSBM where the average degree $d$ goes to infinity. In this limit the SBM is equivalent to a low-rank matrix factorization problem and can be rigorously analyzed. The Bayes-optimality of the belief propagation-based algorithm is then provable.

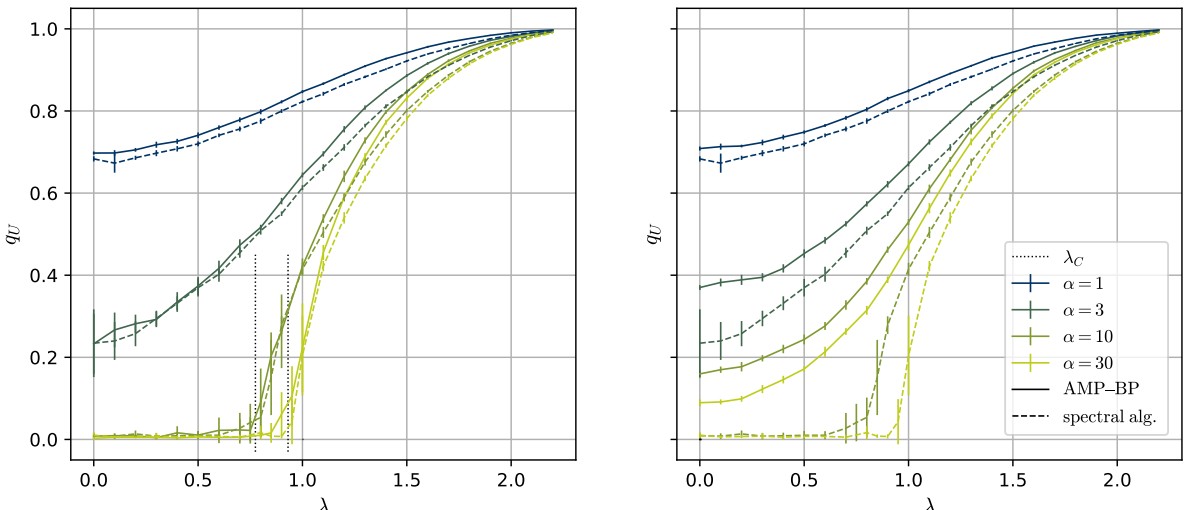

Figure 2: *Performances of AMP–BP and of the spectral algorithm of Deshpande et al. (2018) sec. 4.* Overlap $q_U$ of the fixed point of the algorithms, vs snr $\lambda$ for a range of ratios $\alpha$. *Left:* unsupervised, $\rho = 0$; *right:* semi-supervised, $\rho = 0.1$. Vertical dashed lines on the left: theoretical thresholds $\lambda_c$ to partial recovery, eq. (13). $N = 3 \times 10^4$, $\mu^2 = 4$, $d = 5$. We run ten experiments per point.

The dense limit is defined by $c_i$ and $c_o$ of order $N$ and $c_i - c_o = \sqrt{\nu N}$. The adjacency matrix $A$ can be approximated by a noisy rank-one matrix (Lesieur et al., 2017; Deshpande et al., 2018)

$$A_{ij} \approx \sqrt{\frac{\nu}{N}} u_i u_j + \Xi_{ij} \tag{17}$$

where the $\Xi_{ij}$ are standard independent normals. The CSBM is then a joint *uu* and *uv* matrix factorization problem. The BP on the SBM is approximated by an AMP algorithm; and one can glue the two AMPs. One can prove that the resulting AMP–AMP algorithm is Bayes-optimal.

We state in appendix C the AMP–AMP algorithm for CSBM and provide its state evolution (SE) equations.

The rank-one approximation is valid for average degrees $d$ moderately large. Numerical simulations show that $d \gtrsim 20$ is enough at $N = 10^4$ (Duranthon & Zdeborová, 2023; Shi et al., 2023).

**Parameter estimation and Bethe free entropy** In case the parameters $\theta = (c_i, c_o, \mu)$ of the CSBM are not known they can be estimated using expectation-maximization (EM). This was proposed in Decelle et al. (2011) for the affinity coefficients and the group sizes of the SBM. In the Bayesian framework, one has to find the most probable value of $\theta$. This is equivalent to maximizing the free entropy $\phi$ (8) over $\theta$.

The exact free entropy $\phi$ is not easily computable because this requires integrating over all configurations. It can be computed thanks to AMP–BP: at a fixed point of the algorithm, $\phi$ can be expressed from the values of the variables. It is then called the Bethe free entropy $\phi^{\text{Bethe}}$ in the literature. The derivation is presented in appendix B. $\phi^{\text{Bethe}}$ converges in probability to $\phi$ in the large $N$ limit: for any $\epsilon > 0$, $P(|\phi - \phi^{\text{Bethe}}| > \epsilon) \underset{N \to \infty}{\to} 0$. For compactness, we write $\chi_-^{i \to j} = 1 - \chi_+^{i \to j}$ and pack the connectivity coefficients in the matrix $C$. We have

$$N\phi^{\text{Bethe}} = N\frac{d}{2} + \sum_i \log \sum_u e^{\tilde{h}_u^i} \prod_{k \in \partial i} \sum_t C_{u,t} \chi_t^{k \to i} - \sum_{(ij) \in G} \log \sum_{u,t} C_{u,t} \chi_u^{i \to j} \chi_t^{j \to i} \tag{18}$$

$$+ \sum_\beta \frac{1}{2} \left( \frac{B_U^{\beta,2}}{1 + A_U} - \log(1 + A_U) \right) - \sum_{i,\beta} \left( \sqrt{\frac{\mu}{N}} B_{\beta i} \hat{v}_\beta \hat{u}_i - \frac{\mu}{N} (\frac{1}{2} \hat{v}_\beta^2 + \hat{u}_i^2 \sigma_V - \frac{1}{2} \hat{v}_\beta^2 \hat{u}_i^2) \right) ,$$

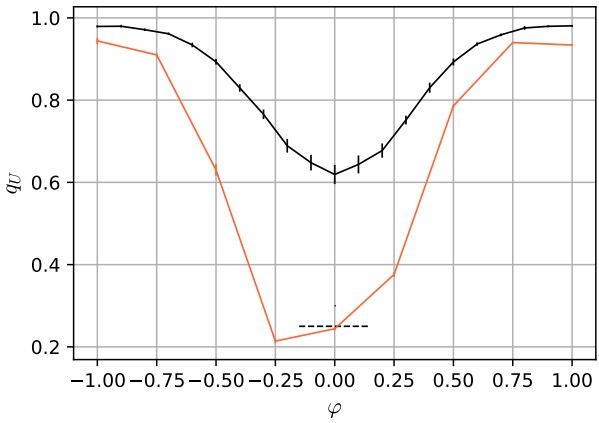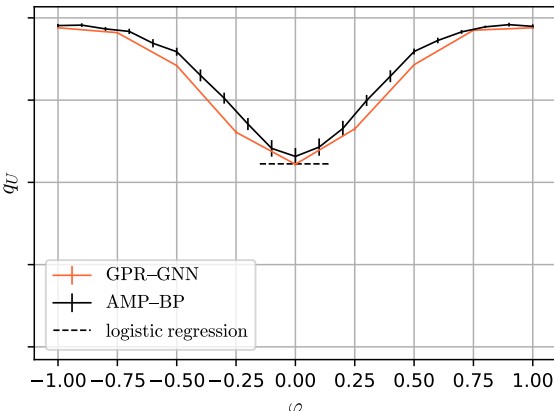

Figure 3: *Comparison against GPR-GNN Chien et al. (2021).* Overlap $q_U$ achieved by the algorithms, vs $\varphi = \frac{2}{\pi} \arctan(\frac{\lambda\sqrt{\alpha}}{\mu})$. *Left:* few nodes revealed $\rho = 0.025$; *right:* more nodes revealed $\rho = 0.6$. For GPR-GNN we plot the results of Fig. 2 and tables 5 and 6 from Chien et al. (2021). $N = 5 \times 10^3$, $\alpha = 2.5$, $\epsilon = 3.25$, $d = 5$. We run ten experiments per point for AMP–BP.

where $\tilde{h}_u^i$, $A_U$ and $B_U^\beta$ are given by the algorithm. One can then estimate the parameters $\theta$ by numerically maximizing $\phi(\theta)^{\text{Bethe}}$, or more efficiently iterating the extremality condition $\nabla_\theta \phi^{\text{Bethe}} = 0$, given in appendix B, which become equivalent to the expectation-maximization algorithm.

The Bethe free entropy is also used to determine the location of a first-order phase transition in case the AMP–BP algorithm has a different fixed point when running from the random initialization as opposed to running from the initialization informed by the ground truth values of the hidden variables $u, v$. In analogy with the standard SBM (Decelle et al., 2011) and the standard high-dimensional Gaussian mixture (Lesieur et al., 2016; 2017), a first-order transition is expected to appear when there are multiple groups or when one of the two groups is much smaller than the other. We only study the case of two balanced groups where we observed these two initializations converge to the same fixed point in all our experiments.

**Semi-supervision and noisy labels**  Semi-supervised AMP–BP can be straightforwardly generalized to deal with noisy labels, thanks to the Bayesian framework we consider. It is sufficient to set the semi-supervised prior $P_{U,i}$ to the actual distribution of the labels. For instance, if the observed label of the train node $i$ is the true $u_i$ with probability $q$ and $-u_i$ otherwise, then $P_{U,i}(s) = q\delta_{s,u_i} + (1 - q)\delta_{s,-u_i}$.

## 4 Comparison against graph neural networks

AMP–BP gives upper bounds for the performance of any other algorithm for solving CSBM. It is thus highly interesting to compare to other algorithms and to see how far from optimality they are.

### 4.1 Comparison to GPR-GNN from previous literature

CSBM has been used as a synthetic benchmark many times (Chien et al., 2021; Cong et al., 2021; Fu et al., 2021; Lei et al., 2022) to assess new architectures of GNNs or new algorithms. These works do not compare their results to optimal performances. We propose to do so. As an illustrative example, we reproduce the experiments from Fig. 2 of the well-known work Chien et al. (2021).

Authors of Chien et al. (2021) proposed a GNN based on a generalized PageRank method; it is called GPR-GNN. The authors test it on CSBM for node classification and show it has better accuracy than many other models, for both $\lambda > 0$ (homophilic graph) and $\lambda < 0$ (heterophilic graph). We reproduce their results in Fig. 3 and compare them to the optimal performance given by AMP–BP. The authors of Chien et al. (2021) use a different parameterization of the CSBM: they consider $\lambda^2 + \mu^2/\alpha = 1 + \epsilon$ and $\varphi = \frac{2}{\pi} \arctan(\frac{\lambda\sqrt{\alpha}}{\mu})$.

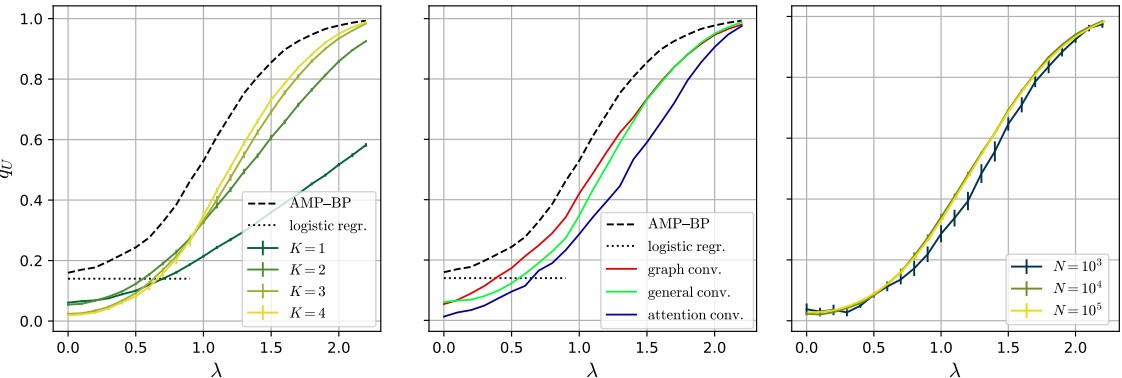

Figure 4: *Comparison to GNNs of various architectures and convergence to a high-dimensional limit.* Overlap $q_U$ achieved by the GNNs, vs the snr $\lambda$. *Left:* general convolution for different numbers of layers $K$; *middle:* for different types of convolutions, at the best $K$ (the detailed results for every $K$ are reported on Fig. 8 of appendix F); *right:* general convolution at $K = 3$ for different sizes $N$. The other parameters are $N = 3 \times 10^4$, $\alpha = 10$, $\mu^2 = 4$, $d = 5$, $\rho = 0.1$. We run five experiments per point.

We see from Fig. 3 that this state-of-the-art GNN can be far from optimality. For the worst parameters in the figure, GPR-GNN reaches an overlap 50% lower than the accuracy of AMP–BP. Fig. 3 left shows that the gap is larger when the training labels are scarce, at $\rho = 2.5\%$. When enough data points are given ($\rho = 60\%$, right), GPR-GNN is rather close to optimality. However, this set of parameters seems easy since at $\varphi = 0$ simple logistic regression is also close to AMP–BP.

Authors of Chien et al. (2021); Fu et al. (2021); Lei et al. (2022) take $\epsilon > 0$ thus considering only parameters in the detectable regime. We argue it is more suitable for unsupervised learning than for semi-supervised because the labels then carry little additional information. From left to right on Fig. 3 we reveal more than one-half of the labels but the optimal performance increases by at most 4%. To have a substantial difference between unsupervised and semi-supervised one should take $\lambda^2 + \mu^2/\alpha < 1$, as we do in Fig. 2. This regime would then be more suitable to assess the learning by empirical risk minimizers (ERMs) such as GNNs. We use this regime in the next section.

### 4.2 Baseline graph neural networks

In this section, we evaluate the performance of a range of baseline GNNs on CSBM. We show again that the GNNs we consider do not reach optimality and that there is room for improving these architectures. We consider the same task as before: on a single instance of CSBM a fraction $\rho$ of node labels are revealed and the GNN must guess the hidden labels. As to the parameters of the CSBM, we work in the regime where supervision is necessary for good inference; i.e. we take $\mu^2/\alpha < 1$.

We use the architectures implemented by the GraphGym package (You et al., 2020). It allows to design the intra-layer and inter-layer architecture of the GNN in a simple and modular manner. The parameters we considered are the number $K$ of message-passing layers, the convolution operation (among graph convolution, general convolution and graph-attention convolution) and the internal dimension $h$. We fixed $h = 64$; we tried higher values for $h$ at $K = 2$, but we observed slight or no differences. One GNN is trained to perform node classification on one instance of CSBM on the whole graph, given the set $\Xi$ of revealed nodes. It is evaluated on the remaining nodes. More details on the architecture and the training are given in appendix E.

Fig. 4 shows that there is a gap between the optimal performance and the one of all the architectures we tested. The GNNs reach an overlap of at least about ten per cent lower than the optimality. They are close to the optimality only near $\lambda = \sqrt{d}$ when the two groups are very well separated. The gap is larger at small $\lambda$. At small $\lambda$ it may be that the GNNs rely too much on the graph while it carries little information: the logistic regression uses only the node features and performs better.

The shown results are close to being asymptotic in the following sense. Since CSBM is a synthetic dataset we can vary $N$, train different GNNs and check whether their test accuracies are the same. Fig 4 right shows that the test accuracies converge to a limit at large $N$ and taking $N = 3 \times 10^4$ is enough to work in this large-size limit of the GNNs on CSBM.

These experiments lead to another finding. We observe that there is an optimal number $K$ of message-passing layers that depends on $\lambda$. Having $K$ too large mixes the covariates of the two groups and diminishes the performance. This effect seems to be mitigated by the attention mechanism: In Fig. 8 right of appendix F the performance of the graph-attention GNN increases with $K$ at every $\lambda$.

It is an interesting question whether the optimum performance can be reached by a GNN. One could argue that AMP–BP is a sophisticated algorithm tailored for this problem, while GNNs are more generic. However, Fig. 4 shows that even logistic regression can be close to optimality at $\lambda = 0$.

We study the effect of the training labels. We consider a setting where $\mu^2/\alpha < 1$ so supervision is necessary for $\lambda < 1$. We check this is the case by letting the training ratio $\rho$ going to 0 on Fig. 9 in appendix F. We observe the resulting accuracies $q_U$ of AMP–BP and the GNNs drop to 0. The transition seems to be sharp. AMP–BP always performs better as expected.

### 4.3  Comparison to Baranwal et al. (2023), a locally Bayes-optimal architecture

The authors of Baranwal et al. (2023) propose a GNN whose architecture implements the locally Bayes-optimal classifier for CSBM. Given a node they consider only its neighborhood; since the graph is sparse, it is tree-like and the likelihood of the node has a simple analytical expression. They parameterize a part of this classifier by a multi-layer perceptron that is trained and a connectivity matrix between communities is also learned. The resulting architecture is a GNN with a specific agregation function. The authors did not name it; we propose the name clipGNN. They set $l$ the size of the neighborhoods it processes and $L$ the number of layers of the perceptron. At $l = 0$ there is no agregation and clipGNN is a multi-layer perceptron classifying a Gaussian mixture.

The setting Baranwal et al. (2023) considers is a bit different: clipGNN was derived in a low-dimensional setting $P = \mathcal{O}(1)$. Yet, for large $P$ it can be run and its local Bayes-optimality should remain valid. For fairness we take $P$ small or not too large. We have to scale $\mu$ the snr of the Gaussian mixture accordingly. According to (13) we shall take $\mu^2$ of order $\alpha = N/P$.

The comparison can be seen on Fig. 5. AMP–BP is considerably better for all parameters we tried. The results are similar to Fig. 4 left in the sense that increasing the neighborhood size $l$ increases the performance of their architecture, but there is still a large gap to reach the performances of AMP–BP. Their architecture works closer to AMP–BP at large $\alpha$ i.e. small $P$.

## 5  Conclusion

We provide the AMP–BP algorithm to solve the balanced CSBM with two groups optimally asymptotically in the limit of large dimension in both the unsupervised and semi-supervised cases. We show a sizable difference between this optimal performance and the one of recently proposed GNNs to which we compare. We hope that future works using CSBM as an artificial dataset will compare to this optimal AMP–BP algorithm, and we expect that this will help in developing more powerful GNN architectures and training methods.

An interesting future direction of work could be to generalize the results of Shi et al. (2023) on the theoretical performance of a one-layer graph-convolution GNN trained on CSBM.

Another promising direction would be unrolling AMP–BP to form a new architecture of GNN, as Chen et al. (2020) did for BP for SBM and Borgerding et al. (2017) for AMP in compressed sensing, and see if it can close the observed algorithmic gap. We expect this new architecture of GNN to be based on non-backtracking walks and to incorporate skip connections between its layers.

### Acknowledgement

We acknowledge funding from the Swiss National Science Foundation grant SMArtNet (grant number 212049).

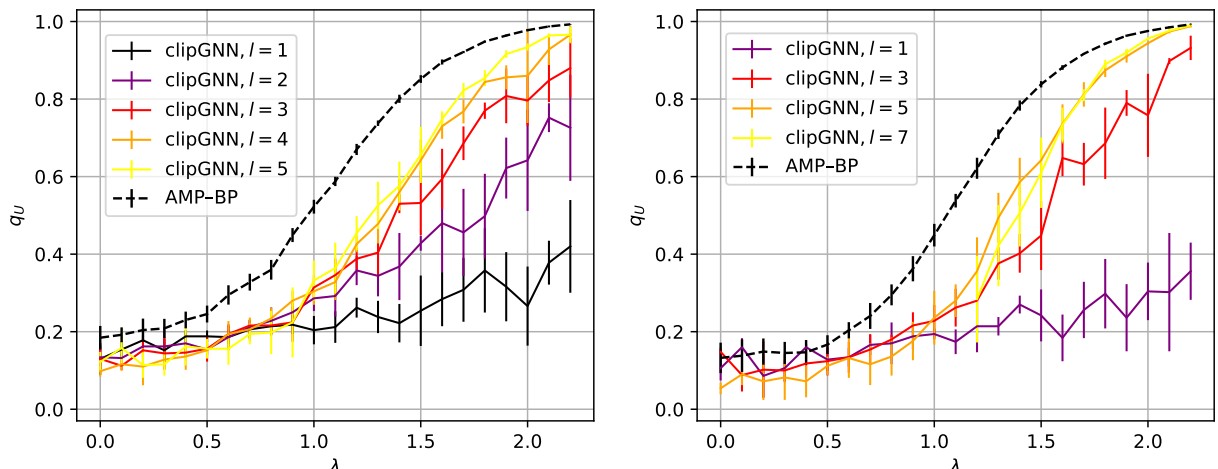

Figure 5: *Comparison against clipGNN (Baranwal et al., 2023).* Overlap $q_U$ achieved by the algorithms, vs $\lambda$. $l$ is the size of the neigborhood clipGNN processes. *Left:* $\mu^2 = 50$ and $\alpha = 50$ i.e. $P = 200$; *right:* $\mu^2 = 500$ and $\alpha = 500$ i.e. $P = 20$. The other parameters are $N = 10^4$ ($N = 5 \times 10^3$ for the two largest $l$), $\rho = 0.05$, $d = 5$, $L = 1$. For clipGNN we run the code kindly provided by the authors; we run five experiments per point.

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

## A Derivation of the algorithm

We recall the setup. We have $N$ nodes $u_i$ in $\{-1, +1\}$, $P$ coordinates $v_\beta$ in $\mathbb{R}$; we are given the $P \times N$ matrix $B_{\beta i} = \sqrt{\frac{\mu}{N}} v_\beta u_i + Z_{\beta i}$, where $Z_{\beta i}$ is standard Gaussian, and we are given a graph whose edges $A_{ki} \in \{0, 1\}$ are drawn according to $P(A_{ki}|u_k, u_i) \propto C_{u_k, u_i}^{A_{ki}} (1 - C_{u_k, u_i}/N)^{1 - A_{ki}}$.

We define $w_{\beta i} = \sqrt{\frac{\mu}{N}} v_\beta u_i$ and $e^{g(B,w)} = e^{-(B-w)^2/2}$ the output channel. Later we approximate the output channel by its expansion near 0; we have: $\frac{\partial g}{\partial w}(w = 0) = B_{\beta i}$ and $\left( \frac{\partial g}{\partial w}(w = 0) \right)^2 + \frac{\partial^2 g}{\partial w^2}(w = 0) = B_{\beta i}^2 - 1$.

We write belief propagation for this problem. We start from the factor graph of the problem:

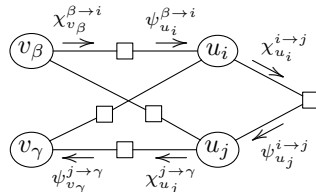

There are six different messages that stem from the factor graph; they are:

$$\chi_{u_i}^{i \to j} \propto P_{U,i}(u_i) \prod_\beta \psi_{u_i}^{\beta \to i} \prod_{k \neq i, j} \psi_{u_i}^{k \to i} \tag{19}$$

$$\psi_{u_j}^{i \to j} \propto \sum_{u_i} \chi_{u_i}^{i \to j} P(A_{ij}|u_i, u_j) \tag{20}$$

$$\chi_{u_i}^{i \to \beta} \propto P_{U,i}(u_i) \prod_{\gamma \neq \beta} \psi_{u_i}^{\gamma \to i} \prod_{k \neq i} \psi_{u_i}^{k \to i} \tag{21}$$

$$\psi_{v_\beta}^{i \to \beta} \propto \sum_{u_i} \chi_{u_i}^{i \to \beta} e^{g(B_{\beta i}, w_{\beta i})} \tag{22}$$

$$\chi_{v_\beta}^{\beta \to i} \propto P_V(v_\beta) \prod_{j \neq i} \psi_{v_\beta}^{j \to \beta} \tag{23}$$

$$\psi_{u_i}^{\beta \to i} \propto \int \mathrm{d}v_\beta \, \chi_{v_\beta}^{\beta \to i} e^{g(B_{\beta i}, w_{\beta i})} \tag{24}$$

where the proportionality sign $\propto$ means up to the normalization factor that insures the message sums to one over its lower index.

We simplify these equations following closely Lesieur et al. (2017) and Decelle et al. (2011).

### A.1 Gaussian mixture part

We parameterize messages 22 and 24 as Gaussians expanding $g$ :

$$\psi_{v_\beta}^{i \to \beta} \propto \sum_{u_i} \chi_{u_i}^{i \to \beta} e^{g(B_{\beta i}, 0)} (1 + B_{\beta i} w_{\beta i} + (B_{\beta i}^2 - 1) w_{\beta i}^2 / 2) \tag{25}$$

$$\psi_{u_i}^{\beta \to i} \propto \int \mathrm{d}v_\beta \, \chi_{v_\beta}^{\beta \to i} e^{g(B_{\beta i}, 0)} (1 + B_{\beta i} w_{\beta i} + (B_{\beta i}^2 - 1) w_{\beta i}^2 / 2) \tag{26}$$

we define

$$\hat{v}_{\beta \to i} = \int \mathrm{d}v \, \chi_v^{\beta \to i} v \quad ; \quad \sigma_V^{\beta \to i} = \int \mathrm{d}v \, \chi_v^{\beta \to i} (v^2 - \hat{v}_{\beta \to i}^2) \tag{27}$$

$$\hat{u}_{i \to \beta} = \sum_u \chi_u^{i \to \beta} u \quad ; \quad \sigma_U^{i \to \beta} = \sum_u \chi_u^{i \to \beta} (u^2 - \hat{u}_{i \to \beta}^2) \tag{28}$$

we assemble products of messages in the target-dependent elements

$$B_V^{i\to\beta} = \sqrt{\frac{\mu}{N}} \sum_{\gamma\neq\beta} B_{\gamma i}\hat{v}_{\gamma\to i} \quad ; \quad B_U^{\beta\to i} = \sqrt{\frac{\mu}{N}} \sum_{j\neq i} B_{\beta j}\hat{u}_{j\to\beta} \tag{29}$$

$$A_V^{i\to\beta} = \frac{\mu}{N} \sum_{\gamma\neq\beta} B_{\gamma i}^2 \hat{v}_{\gamma\to i}^2 - (B_{\gamma i}^2 - 1)(\hat{v}_{\gamma\to i}^2 + \sigma_V^{\gamma\to i}) \tag{30}$$

$$A_U^{\beta\to i} = \frac{\mu}{N} \sum_{j\neq i} B_{\beta j}^2 \hat{u}_{j\to\beta}^2 - (B_{\beta j}^2 - 1)(\hat{u}_{j\to\beta}^2 + \sigma_U^{j\to\beta}) \tag{31}$$

and in the target-independent elements

$$B_V^i = \sqrt{\frac{\mu}{N}} \sum_{\beta} B_{\beta i}\hat{v}_{\beta\to i} \quad ; \quad B_U^\beta = \sqrt{\frac{\mu}{N}} \sum_j B_{\beta j}\hat{u}_{j\to\beta} \tag{32}$$

$$A_V^i = \frac{\mu}{N} \sum_{\beta} B_{\beta i}^2 \hat{v}_{\beta\to i}^2 - (B_{\beta i}^2 - 1)(\hat{v}_{\beta\to i}^2 + \sigma_V^{\beta\to i}) \tag{33}$$

$$A_U^\beta = \frac{\mu}{N} \sum_j B_{\beta j}^2 \hat{u}_{j\to\beta}^2 - (B_{\beta j}^2 - 1)(\hat{u}_{j\to\beta}^2 + \sigma_U^{j\to\beta}) \tag{34}$$

so we can write the messages of eq. 19, 21 and 23 in a close form as

$$\chi_{u_i}^{i\to j} \propto P_{U,i}(u_i)e^{u_i B_V^i - u_i^2 A_V^i/2} \prod_{k\neq i,j} \sum_{u_k} \chi_{u_k}^{k\to i} P(A_{ki}|u_k, u_i) \tag{35}$$

$$\chi_{u_i}^{i\to\beta} \propto P_{U,i}(u_i)e^{u_i B_V^{i\to\beta} - u_i^2 A_V^{i\to\beta}/2} \prod_{k\neq i} \sum_{u_k} \chi_{u_k}^{k\to i} P(A_{ki}|u_k, u_i) \tag{36}$$

$$\chi_{v_\beta}^{\beta\to i} \propto P_V(v_\beta)e^{v_\beta B_U^{\beta\to i} - v_\beta^2 A_U^{\beta\to i}/2} \tag{37}$$

Since we sum over $u = \pm 1$, the $A_V$s can be absorbed in the normalization factor and we can omit them.

## A.2 SBM part

We work out the SBM part using standard simplifications. We define the marginals and their fields by

$$\chi_{u_i}^i \propto P_{U,i}(u_i)e^{u_i B_V^i} \prod_{k\neq i} \sum_{u_k} \chi_{u_k}^{k\to i} P(A_{ki}|u_k, u_i) \tag{38}$$

$$h_{u_i} = \frac{1}{N} \sum_k \sum_{u_k} C_{u_k, u_i} \chi_{u_k}^k \tag{39}$$

Simplifications give

$$\chi_{u_i}^{i\to j} = \chi_{u_i}^i \quad \text{if } (ij) \notin G \quad ; \quad \text{else} \tag{40}$$

$$\chi_{u_i}^{i\to j} \propto P_{U,i}(u_i)e^{u_i B_V^i}e^{-h_{u_i}} \prod_{k\in\partial i/j} \sum_{u_k} C_{u_k, u_i} \chi_{u_k}^{k\to i} \tag{41}$$

$$\chi_{u_i}^i \propto P_{U,i}(u_i)e^{u_i B_V^i}e^{-h_{u_i}} \prod_{k\in\partial i} \sum_{u_k} C_{u_k, u_i} \chi_{u_k}^{k\to i} \tag{42}$$

$$\chi_{u_i}^{i\to\beta} \propto P_{U,i}(u_i)e^{u_i B_V^{i\to\beta}}e^{-h_{u_i}} \prod_{k\in\partial i} \sum_{u_k} C_{u_k, u_i} \chi_{u_k}^{k\to i} \tag{43}$$

### A.3 Update functions

The estimators can be updated thanks to the functions

$$f_V(A, B) = \frac{\int \mathrm{d}v \, v P_V(v) \exp\left(Bv - Av^2/2\right)}{\int \mathrm{d}v \, P_V(v) \exp\left(Bv - Av^2/2\right)} = B/(A+1) \tag{44}$$

$$f_U(A, B, \chi) = \frac{\sum_u u P_{U,i}(u) \exp\left(Bu\right) \chi_u}{\sum_u P_{U,i}(u) \exp\left(Bu\right) \chi_u} \tag{45}$$

$$\partial_B f_U = 1 - f_U^2 \tag{46}$$

The update is

$$\hat{v}_{\beta \to i} = f_V(A_U^{\beta \to i}, B_U^{\beta \to i}) \quad ; \quad \sigma_V^{\beta \to i} = \partial_B f_V(A_U^{\beta \to i}, B_U^{\beta \to i}) \tag{47}$$

$$\hat{u}_{i \to \beta} = f_U(A_V^{i \to \beta}, B_V^{i \to \beta}, \hat{\chi}^i) \quad ; \quad \sigma_U^{i \to \beta} = \partial_B f_U(A_V^{i \to \beta}, B_V^{i \to \beta}, \hat{\chi}^i) \tag{48}$$

where $\hat{\chi}_u^i = e^{-h_u} \prod_{k \in \partial i} \sum_{u_k} C_{u_k, u} \chi_{u_k}^{k \to i}$.

### A.4 Time indices

We mix the AMP part and the BP part in this manner:

$$\hat{u}, \sigma_U^{(t)} \longrightarrow A_U, B_U^{(t+1)}; \hat{v}, \sigma_V^{(t+1)} \longrightarrow A_V, B_V^{(t+1)} \longrightarrow \hat{u}, \sigma_U^{(t+1)}$$

$$\mid \quad \mid \simeq \qquad\qquad\qquad\qquad\qquad\qquad\qquad\qquad \mid \simeq$$

$$\chi^{(t)} \longrightarrow \hat{\chi}, h^{(t+1)} \longrightarrow \chi^{(t+1)}$$

where the dashed lines mean that $\hat{u}_{i \to \beta}$ and $\chi^i$ are close. We precise this statement in the next section.

### A.5 Additional simplifications preserving asymptotic accuracy

We introduce the target-independent estimators

$$\hat{v}_\beta = f_V(A_U^\beta, B_U^\beta) \quad ; \quad \sigma_V^\beta = \partial_B f_V(A_U^\beta, B_U^\beta) \tag{49}$$

$$\hat{u}_i = f_U(A_V^i, B_V^i, \hat{\chi}^i) = \sum_u u \chi_u^i \quad ; \quad \sigma_U^i = \partial_B f_U(A_V^i, B_V^i, \hat{\chi}^i) \tag{50}$$

This makes the message of eq. 43 redundant: we can directly express $\hat{u}^i$, the estimator of the AMP side, as a simple function of $\chi_u^i$, the estimator of the BP side.

We express the target-independent $A$s and $B$s as a function of these. We evaluate the difference between the target-independent and the target-dependent estimators and we obtain

$$A_V^{i,(t+1)} = A_V^{i \to \beta,(t+1)} \quad ; \quad A_U^{\beta,(t+1)} = A_U^{\beta \to i,(t+1)} \tag{51}$$

$$B_V^{i,(t+1)} = \sqrt{\frac{\mu}{N}} \sum_\beta B_{\beta i} \hat{v}_\beta^{(t+1)} - \frac{\mu}{N} \sum_\beta B_{\beta i}^2 \sigma_V^{\beta,(t+1)} \hat{u}_i^{(t)} \tag{52}$$

$$B_U^{\beta,(t+1)} = \sqrt{\frac{\mu}{N}} \sum_i B_{\beta i} \hat{u}_i^{(t)} - \frac{\mu}{N} \sum_i B_{\beta i}^2 \sigma_U^{i,(t)} \hat{v}_\beta^{(t)} \tag{53}$$

We further notice that $B_{\beta i}^2$ concentrate on one; this simplifies the equations to

$$A_V^{(t+1)} = \frac{\mu}{N} \sum_\beta \hat{v}_\beta^{(t+1),2} \quad ; \quad A_U^{(t+1)} = \frac{\mu}{N} \sum_i \hat{u}_i^{(t+1),2} \tag{54}$$

$$B_V^{i,(t+1)} = \sqrt{\frac{\mu}{N}} \sum_\beta B_{\beta i} \hat{v}_\beta^{(t+1)} - \frac{\mu}{N} \sum_\beta \sigma_V^{\beta,(t+1)} \hat{u}_i^{(t)} \tag{55}$$

$$B_U^{\beta,(t+1)} = \sqrt{\frac{\mu}{N}} \sum_i B_{\beta i} \hat{u}_i^{(t)} - \frac{\mu}{N} \sum_i \sigma_U^{i,(t)} \hat{v}_\beta^{(t)} \tag{56}$$

The $A$s do not depend on the node then; this simplifies $\sigma_V$:

$$\sigma_V^{(t+1)} = 1/(1 + A_U^{(t+1)}) \tag{57}$$

$$\hat{v}_\beta^{(t+1)} = \sigma_V^{(t+1)} B_U^{\beta,(t+1)} \tag{58}$$

$$B_V^{i,(t+1)} = \sqrt{\frac{\mu}{N}} \sum_\beta B_{\beta i} \hat{v}_\beta^{(t+1)} - \frac{\mu}{\alpha} \sigma_V^{(t+1)} \hat{u}_i^{(t)} \tag{59}$$

Last, we express all the updates in function of $\chi_{u=+1}$, having $\chi_{u=-1} = 1 - \chi_{u=+1}$. This gives the algorithm in the main part.

## B   Free entropy and estimation of the parameters

To compute Bethe free entropy we start from the factor graph. Factor nodes are between two variables so the free entropy is

$$N\phi = \sum_i \phi_i - \sum_{i<j} \phi_{(ij)} + \sum_\beta \phi_\beta - \sum_{(i\beta)} \phi_{(i\beta)} \tag{60}$$

with

$$\phi_i = \log \sum_{u_i} P_{U,i}(u_i) \prod_{i\neq j} \psi_{u_i}^{j\to i} \prod_\beta \psi_{u_i}^{\beta\to i} \tag{61}$$

$$\phi_{(ij)} = \log \sum_{u_i,u_j} P(A_{ij}|u_i,u_j) \chi_{u_i}^{i\to j} \chi_{u_j}^{j\to i} \tag{62}$$

$$\phi_\beta = \log \int \mathrm{d}v_\beta \, P_V(v_\beta) \prod_i \psi_{v_\beta}^{i\to\beta} \tag{63}$$

$$\phi_{(i\beta)} = \log \sum_{u_i} \int \mathrm{d}v_\beta \, e^{g(B_{\beta i},w_{\beta i})} \chi_{u_i}^{i\to\beta} \chi_{v_\beta}^{\beta\to i} \tag{64}$$

We use the same simplification as above to express these quantities in terms of the estimators returned by AMP–BP. This is standard computation; we follow Lesieur et al. (2017) and Decelle et al. (2011). The parts $\phi_i$ and $\phi_\beta$ on the variables involves the normalization factors of the marginals $\hat{u}_i$ and $\hat{v}_\beta$:

$$\phi_i = -\frac{A_V}{2} + \log \sum_{u=\pm 1} e^{\hat{h}_u^i} \prod_{k\in\partial i} \sum_{t=\pm 1} C_{u,t} \chi_u^{k\to i} \tag{65}$$

$$\phi_\beta = \log \int \mathrm{d}v \, P_V(v) e^{B_U^\beta v - A_U v^2/2} = \frac{1}{2}\left(\frac{B_U^{\beta,2}}{1+A_U} - \log(1+A_U)\right) \tag{66}$$

where as before

$$\tilde{h}_u^i = -h_u + \log P_{U,i}(u) + uB_V^i \tag{67}$$

$$h_u = \frac{1}{N} \sum_i \sum_{t=\pm 1} C_{u,t} \frac{1 + t\hat{u}_i}{2} \tag{68}$$

$$B_V^i = \sqrt{\frac{\mu}{N}} \sum_\beta B_{\beta i} \hat{v}_\beta - \frac{\mu}{\alpha} \sigma_V \hat{u}_i \tag{69}$$

$$B_U^\beta = \sqrt{\frac{\mu}{N}} \sum_i B_{\beta i} \hat{u}_i - \frac{\mu}{N} \sum_i \sigma_U^i \hat{v}_\beta \tag{70}$$

$$A_V = \frac{\mu}{N} \sum_\beta \hat{v}_\beta^2 \quad ; \quad A_U = \frac{\mu}{N} \sum_i \hat{u}_i^2 \tag{71}$$

$$\tag{72}$$

Then, the edge contributions can be expressed using standard simplifications for SBM:

$$\sum_{i<j} \phi_{(ij)} = \sum_{(ij)\in G} \log \sum_{u,t} C_{u,t} \chi_u^{i\to j} \chi_t^{j\to i} - N\frac{d}{2} \tag{73}$$

For the Gaussian mixture side, we use the same approximations as before, expanding in $w$, integrating over the messages and simplifying. We remove the constant part $g(B, 0)$ to obtain

$$\phi_{(i\beta)} = \sqrt{\frac{\mu}{N}} B_{\beta i} \hat{v}_\beta \hat{u}_i - \frac{\mu}{N} (\hat{v}_\beta^2 \sigma_U^i + \hat{u}_i^2 \sigma_V + \frac{1}{2} \hat{v}_\beta^2 \hat{u}_i^2) \tag{74}$$

Last we replace $A_V$ by its expression and $\sigma_U^i = 1 - \hat{u}_i^2$; we assemble the previous equations and we obtain

$$N\phi = N\frac{d}{2} + \sum_i \log \sum_u e^{\tilde{h}_u^i} \prod_{k\in\partial i} \sum_t C_{u,t} \chi_t^{k\to i} - \sum_{(ij)\in G} \log \sum_{u,t} C_{u,t} \chi_u^{i\to j} \chi_t^{j\to i} \tag{75}$$

$$+ \sum_\beta \frac{1}{2} \left( \frac{B_U^{\beta,2}}{1 + A_U} - \log(1 + A_U) \right) - \sum_{i,\beta} \left( \sqrt{\frac{\mu}{N}} B_{\beta i} \hat{v}_\beta \hat{u}_i - \frac{\mu}{N}(\frac{1}{2}\hat{v}_\beta^2 + \hat{u}_i^2 \sigma_V - \frac{1}{2}\hat{v}_\beta^2 \hat{u}_i^2) \right)$$

**Parameter estimation** In case the parameters $\theta = (c_i, c_o, \mu)$ of the CSBM are not known, their actual values are those that maximize the free entropy. This must be understood in this manner: we generate an instance of CSBM with parameters $\theta^*$; we compute the fixed point of AMP–BP at $\theta$ and compute $\phi(\theta)$; then $\phi$ is maximal at $\theta = \theta^*$.

One can find $\theta^*$ thanks to grid search and gradient ascent on $\phi$. We compute the gradient of the free entropy $\phi$ with respect to the parameters $(c_i, c_o, \mu)$. This requires some care: at the fixed point, the messages (i.e. $\chi$, $\hat{u}$, $\hat{v}$ and $\sigma_V$) extremize $\phi$ and therefore its derivative with respect to them is null. We have

$$\partial_{c_i} \phi = -\frac{1}{4} + \frac{1}{N} \sum_{(ij)\in G} \frac{\sum_u \chi_u^{i\to j} \chi_u^{j\to i}}{\sum_{u,t} C_{u,t} \chi_u^{i\to j} \chi_t^{j\to i}} \tag{76}$$

$$\partial_{c_o} \phi = -\frac{1}{4} + \frac{1}{N} \sum_{(ij)\in G} \frac{\sum_u \chi_u^{i\to j} \chi_{-u}^{j\to i}}{\sum_{u,t} C_{u,t} \chi_u^{i\to j} \chi_t^{j\to i}} \tag{77}$$

$$\partial_\mu \phi = \frac{1}{2N} \left( \frac{1}{\sqrt{\mu N}} \sum_{i,\beta} B_{\beta i} \hat{v}_\beta \hat{u}_i - \sum_\beta \hat{v}_\beta^2 - \frac{1}{\alpha} \sigma_V \sum_i \hat{u}_i^2 \right) \tag{78}$$

We emphasize that in these equations the messages are the fixed point of AMP–BP run at $(c_i, c_o, \mu)$. At each iteration one has to run again AMP–BP with the new estimate of the parameters.

A clever update rule is possible. We equate the gradient of $\phi$ to zero and obtain that:

$$c_i = \frac{4}{N} \sum_{(ij) \in G} \frac{\sum_u C_{u,u} \chi_u^{i \to j} \chi_u^{j \to i}}{\sum_{u,t} C_{u,t} \chi_u^{i \to j} \chi_t^{j \to i}} \tag{79}$$

$$c_o = \frac{4}{N} \sum_{(ij) \in G} \frac{\sum_u C_{u,-u} \chi_u^{i \to j} \chi_{-u}^{j \to i}}{\sum_{u,t} C_{u,t} \chi_u^{i \to j} \chi_t^{j \to i}} \tag{80}$$

$$\mu = \left( \frac{\frac{\alpha}{\sqrt{N}} \sum_{i,\beta} B_{\beta i} \hat{v}_\beta \hat{u}_i}{\alpha \sum_\beta \hat{v}_\beta^2 + \sigma_V \sum_i \hat{u}_i^2} \right)^2 \tag{81}$$

These equations can be interpreted as the update of a maximization-expectation algorithm: we enforce the parameters to be equal to the value estimated by AMP–BP.

We remark that these updates are those of standard SBM and Gaussian mixture. The difference with CSBM appears only implicitly in the fixed-point messages.

## C  Dense limit

We state the dense limit of the CSBM and the associated belief propagation-based algorithm.

The dense limit is defined by an average degree $d = (c_i + c_o)/2 \approx c_o$ of order $N$ and $c_i - c_o = \nu \sqrt{N}$. The adjacency matrix $A$ can be approximated by a noisy rank-one matrix whose noise is related to the parameters of the SBM (Lesieur et al., 2017; Deshpande et al., 2018). The dense CSBM is a joint low-rank matrix factorization problem.

One can write belief propagation for this composed problem and simplify them as done in part A. The resulting algorithm is made of two AMP parts that exchange messages. We call it AMP–AMP.

To state the algorithm we need to introduce a transformed adjacency matrix and an inverse noise

$$S_{ij} = \frac{1}{2} \left( \frac{A_{ij}}{\tilde{d}} - \frac{1 - A_{ij}}{1 - \tilde{d}} \right) \quad ; \quad \Delta_I = \frac{\nu^2}{4\tilde{d}(1 - \tilde{d})} \tag{82}$$

where $\tilde{d} = d/N = \mathcal{O}(1)$ and $A$ being the binary adjacency matrix. We define the input function

$$f_U(B, P_U) = \frac{\sum_{u = \pm 1} P_U(u) u e^{uB}}{\sum_{u = \pm 1} P_U(u) e^{uB}} \ . \tag{83}$$

The algorithm then stated in Algorithm 2.

The performance of this algorithm can be tracked by a few scalar equations called state evolution (SE) equations. We define the order parameters

$$m_u^t = \frac{1}{N} \sum_i \hat{u}_i^{(t)} u_i \quad ; \quad m_v^t = \frac{1}{M} \sum_\beta \hat{v}_\beta^{(t)} v_\beta \ . \tag{84}$$

These are the overlaps (or magnetizations) of the estimates at step $t$ with respect to the ground truth. The SE for AMP–AMP on CSBM read

$$m^t = \frac{\mu}{\alpha} m_v^t + \Delta_I m_u^{t-1} \tag{85}$$

$$m_u^t = \rho + (1 - \rho) \mathbb{E}_{u^0, W} \left[ \tanh \left( m^t u^0 + \sqrt{m^t} W \right) u^0 \right] \tag{86}$$

$$m_v^{t+1} = \frac{\mu m_u^t}{1 + \mu m_u^t} \tag{87}$$

where $u^0$ is Rademacher and $W$ is a standard scalar Gaussian.

On can prove rigorously that AMP–AMP follows these SE equations, establish its performances and prove that they are Bayes-optimal.

---

**Input:** features $B_{\beta i} \in \mathbb{R}^{P \times N}$, transformed adjacency matrix $S$, prior information $P_{U,i}$.

**Initialization:** $\hat{u}_i^{(0)} = 2P_{U,i}(+) - 1 + \epsilon^i$, $\hat{v}_\beta^{(0)} = \epsilon^\beta$, $t = 0$, where $\epsilon^i$ and $\epsilon^\beta$ are small centered random variables in $\mathbb{R}$.

**Repeat until convergence:**

$$\sigma_U^i = 1 - \hat{u}_i^{(t),2}$$

AMP estimation of $\hat{v}$

$$A_U = \frac{\mu}{N} \sum_i \hat{u}_i^{(t),2}$$

$$B_U^\beta = \sqrt{\frac{\mu}{N}} \sum_i B_{\beta i} \hat{u}_i^{(t)} - \frac{\mu}{N} \sum_i \sigma_U^i \hat{v}_\beta^{(t)}$$

$$\hat{v}_\beta^{(t+1)} \leftarrow B_U^\beta / (1 + A_U)$$

$$\sigma_V = 1/(1 + A_U)$$

AMP estimation of $\hat{u}$

$$B_V^i = \sqrt{\frac{\mu}{N}} \sum_\beta B_{\beta i} \hat{v}_\beta^{(t+1)} - \frac{\mu}{\alpha} \sigma_V \hat{u}_i^{(t)}$$

$$B_{UU}^i = \frac{\nu}{\sqrt{N}} \sum_{k \neq i} S_{ki} \hat{u}_k^{(t)} - \frac{\Delta_I}{N} \hat{u}_i^{(t-1)} \sum_k \sigma_U^k$$

$$\hat{u}_i^{(t+1)} = f_U(B_{UU}^i + B_V^i, P_{U,i})$$

Update time $t \leftarrow t + 1$.
**Output:** estimated groups $\mathrm{sign}(\hat{u}_i)$.

**Algorithm 2:** *The AMP–AMP algorithm.*

## D    Multiple unbalanced communities

We state a more general formulation of AMP–BP in case there are multiple communities.

We consider $r = \mathcal{O}(1)$ unbalanced groups; they are encoded by the canonical base vectors of $\mathbb{R}^r$ i.e., for $i = 1 \ldots N$, $u_i \in \{(1,0,0,\ldots),(0,1,0,\ldots),\ldots\}$. We assume they are drawn independantly according to $P_U$ for all $i$.

We consider a symmetric connectivity matrix $C \in \mathbb{R}^{r \times r}$ and the graph is generated by

$$P(A_{ij} = 1 | u_i, u_j) = C_{u_i, u_j}/N \tag{88}$$

where we used the notation $C_{u_i, u_j}$ for $u_i^T C u_j$. The features are

$$B_{\beta i} = \sqrt{\frac{\mu}{N}} v_\beta^T u_i + Z_{\beta i} \tag{89}$$

for $\beta = 1 \ldots P$, where $v_\beta \sim \mathcal{N}(0, Id_r)$ determine the randomly drawn centroids and $Z_{\beta i}$ is standard Gaussian noise. The semi-supervised prior is

$$P_{U,i}(s) = \left\{ \begin{array}{ll} \delta_{s,u_i} & \text{if} \quad i \in \Xi, \\ P_U(s) & \text{if} \quad i \notin \Xi. \end{array} \right. \tag{90}$$

We work in the high-dimensional regime $N \to \infty$, $N/P = \alpha = \mathcal{O}(1)$, $\mu = \mathcal{O}(1)$ and the coefficients of $C$ of order one.

We derive the resulting AMP–BP algorithm following Decelle et al. (2011) on the SBM side and Lesieur et al. (2016) on the high-dimensional Gaussian mixture side. One needs to merge these two algorithms along the same lines we did for the case $r = 2$. In the following we consider variables $\hat{u}_i$ and $\hat{v}_\beta$ in $\mathbb{R}^r$. The resulting algorithm is stated in Algorithm 3.

## E    Details on numerical simulations

To define and train the GNNs we use the package provided by You et al. (2020).[2] We implemented the generation of the CSBM dataset.

---

[2]https://github.com/snap-stanford/GraphGym/tree/daded21169ec92fde8b1252b439a8fac35b07d79

**Input:** features $B$, adjacency matrix A, affinity matrix $C$, prior information $P_{U,i}$.

**Initialization:** for $(ij) \in G$, $\chi_{u_i}^{i \to j,(0)} = \epsilon_{u_i}^{i \to j} + P_{U,i}(u_i)$, $\hat{u}_i^{(0)} = \sum_s s P_{U,i}(s)$, $\hat{v}_\alpha^{(0)} = \epsilon_\alpha$, $t = 0$, where $\epsilon^{i \to j}$ and $\epsilon_\alpha$ are zero-mean small random variables in $\mathbb{R}^r$.

**Repeat until convergence:**

$$\sigma_U^i = \mathrm{diag}(\hat{u}_i^{(t)}) - \hat{u}_i^{(t)} \hat{u}_i^{T,(t)}$$

$\mathrm{diag}(s)$ being the $r \times r$ diagonal matrix filled with $s$. AMP update of $\hat{v}$

$$A_U = \frac{\mu}{N} \sum_i \hat{u}_i^{(t)} \hat{u}_i^{T,(t)}$$

$$B_U^\alpha = \sqrt{\frac{\mu}{N}} \sum_i B_{\alpha i} \hat{u}_i^{(t)} - \frac{\mu}{N} \sum_i \sigma_U^i \hat{v}_\alpha^{(t)}$$

$$\hat{v}_\alpha^{(t+1)} \leftarrow (I_r + A_U)^{-1} B_U^\alpha$$

$$\sigma_V = (I_r + A_U)^{-1}$$

AMP estimation of $\hat{u}$

$$A_V = \frac{\mu}{N} \sum_\beta \hat{v}_\beta^{(t)} \hat{v}_\beta^{(t),T}$$

$$B_V^i = \sqrt{\frac{\mu}{N}} \sum_\beta B_{\beta i} \hat{v}_\beta^{(t+1)} - \frac{\mu}{\alpha} \sigma_V \hat{u}_i^{(t)}$$

Estimation of the field $h$

$$h_s = \frac{1}{N} \sum_i \sum_{u_i} C_{s,u_i} \chi_{u_i}^{i,(t)}$$

BP update of the messages $\chi^{i \to j}$ for $(ij) \in G$ and of marginals $\chi^i$

$$\chi_s^{i \to j,(t+1)} \leftarrow \frac{P_{U,i}(s)}{Z^{i \to j}} e^{-h_s + s^T B_V^i - s^T A_V s/2}$$
$$\prod_{k \in \partial i \setminus j} \sum_{u_k} C_{u_k,s} \chi_{u_k}^{k \to i,(t)}$$

$$\chi_s^i = \frac{P_{U,i}(s)}{Z^i} e^{-h_s + s^T B_V^i - s^T A_V s/2}$$
$$\prod_{k \in \partial i} \sum_{u_k} C_{u_k,s} \chi_{u_k}^{k \to i,(t)}$$

$Z^{i \to j}$ and $Z^i$ being normalization factors.
BP estimation of $\hat{u}$

$$\hat{u}_i^{(t+1)} = \sum_{u_i} u_i \chi_{u_i}^i$$

Update time $t \leftarrow t + 1$.
**Output:** estimated means $\hat{u}_i$ and $\hat{v}_\alpha$.

**Algorithm 3:** *The multi-community AMP–BP algorithm.*

**Intra-layer parameters:** we take the internal dimension $h = 64$; we use batch normalization; no dropout; PReLU activation; add aggregation; convolution operation in {generalconv, gcnconv, gatconv} (as defined in the config.py file).

**Inter-layer design:** we take $K \in \{1, 2, 3, 4\}$ layers of message-passing; no pre-process layer; one post-process layer; stack connection.

**Training configuration:** The batch size is one, we train on the entire graph, revealing a proportion $\rho$ of labels; the learning rate is $3 \times 10^{-3}$; we train for forty epochs with Adam; weight decay is $5 \times 10^{-4}$.

For each experiment, we run five independent simulations and report the average of the accuracies at the best epochs.

For the logistic regression, we consider only $\lambda = 0$. We train using gradient descent. We use L2 regularization over the weights and we optimize over its strength.

## F    Supplementary figures

### F.1    The AMP–BP algorithm

We provide experimental evidence showing that for AMP–BP the number of steps to converge does not depend on $N$ and is of order ten.

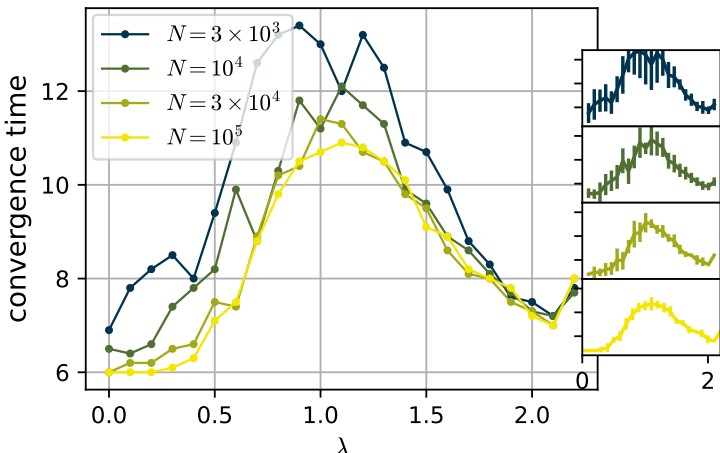

Figure 6:    *Convergence time of AMP–BP.* Average number of iterations for AMP–BP to converge. Convergence is achieved when the overlap $q_U$ varies by less than $10^{-3}$ between two iterations. $\alpha = 10$, $\mu^2 = 4$, $\rho = 0.1$, $d = 5$. We run ten experiments per point.

We check the correctness of the AMP–BP we derived with Monte-Carlo simulations. They give very close results.

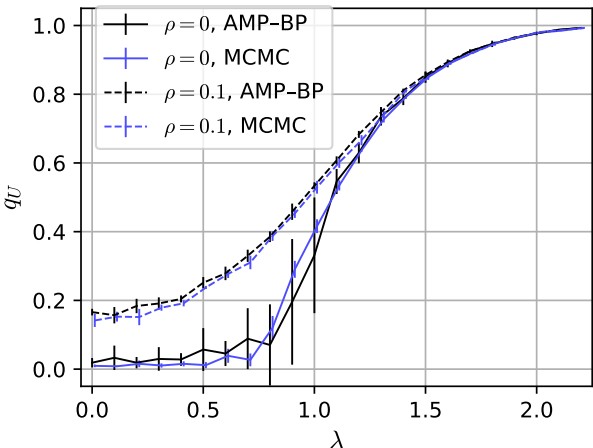

Figure 7:    *Monte-Carlo simulation.* Overlap $q_U$ obtained by AMP–BP and by sampling the posterior (7) thanks to the Metropolis algorithm (MCMC), vs $\lambda$. $N = 10^4$, $\alpha = 10$, $\mu^2 = 4$, $d = 5$. We run five experiments per point.

### F.2 Comparison against graph neural networks

We compare the performance of a range of baselines GNNs to the optimal performances on CSBM.

We report the results of section 4.2 for two supplementary types of convolution. The experiment is the same as the one illustrated by Fig. 4 left, where we train a GNN on CSBM for different number $K$ of layers at many snrs $\lambda$. Fig. 8 is summarized in Fig. 4 middle, where we consider only the best $K$ at each $\lambda$.

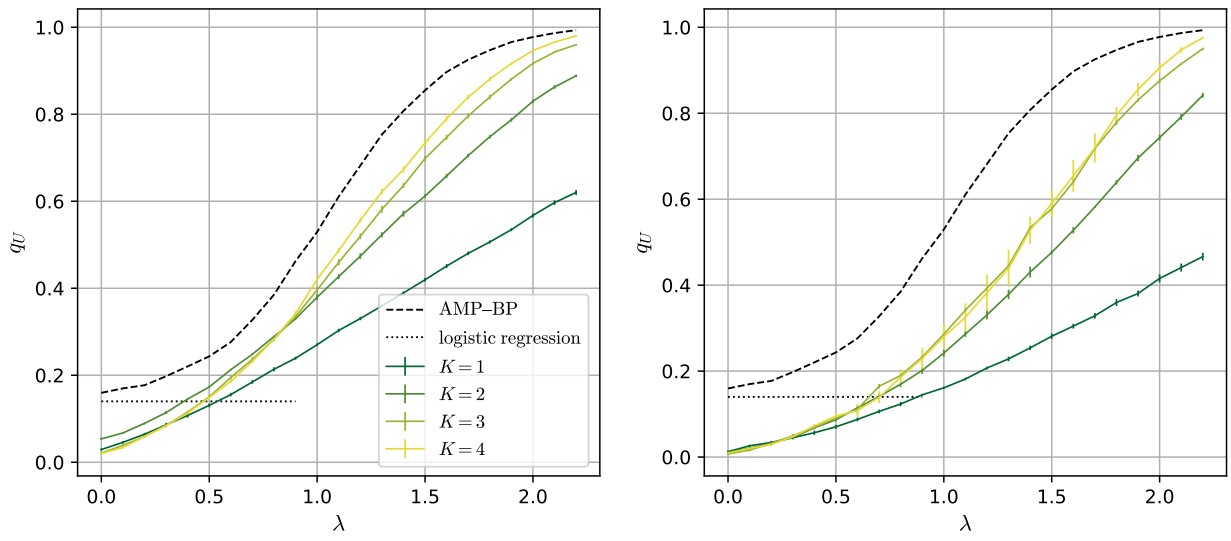

Figure 8: *Comparison to GNNs of various architectures.* Overlap $q_U$ achieved by the GNNs, vs the snr $\lambda$ for different numbers of layers $K$. *Left:* graph convolution; *right:* graph-attention convolution. The other parameters are $N = 3 \times 10^4$, $\alpha = 10$, $\mu^2 = 4$, $d = 5$, $\rho = 0.1$. We run five experiments per point.

We let $\rho$ going to zero to observe the effect of the training labels and how the accuracy diminishes.

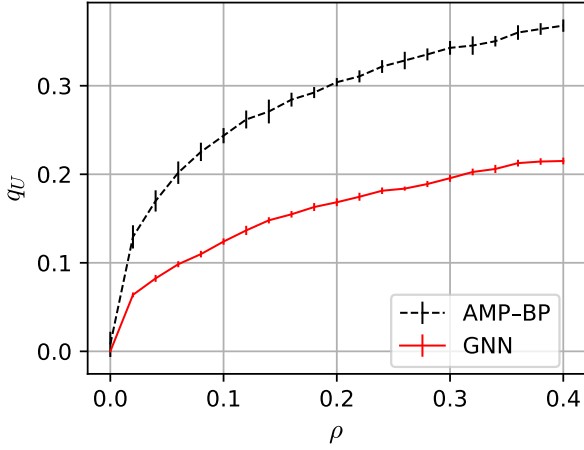

Figure 9: *Effect of training labels.* Overlap $q_U$ vs $\rho$, for AMP–BP and a GNN (general convolution, $K = 2$). The other parameters are $N = 3 \times 10^4$, $\lambda = 0.5$, $\alpha = 10$, $\mu^2 = 4$, $d = 5$. We run ten experiments per point for AMP–BP, five for the GNN.

