# OpenReview forum: "Optimal Inference in Contextual Stochastic Block Models"
_TMLR — Accepted by TMLR_

### Review · Reviewer_hkt1 · 2023-12-31

**Summary Of Contributions:**

The authors study the algorithm, AMP-BP, which is conjectured to achieve the optimal node classification accuracy on the contextual stochastic blockmodel (cSBM). The method is extensively tested and compared with graph neural networks (GNNs), which has demonstrated the performance gap between existing GNNs to a potential optimal performance for cSBM. A simple-to-use implementation is also provided.

**Audience:**

Yes

**Claims And Evidence:**

Yes

**Requested Changes:**

It would be great if the two (especially the last) weaknesses could be addressed. Yet it is also fine in its current form.

**Strengths And Weaknesses:**

Strengths:

(+) Great literature review, which clearly demonstrated the contribution of this work on top of existing literature.

(+) Extensive experiments and comparison with GNNs, including those specialized to cSBM such as Baranwal et al. (2023).

(+) The key message that existing GNN is still sub-optimal, at least for cSBM.

(+) The paper is very well-written.

Weaknesses:

I think this work is great and the weaknesses below should not be used as an objection to its acceptance.

(-) While well-motivated, the optimality of the proposed algorithm is not proven.

(-) The work raises the problem of "GNN is sub-optimal" by showing that AMP-BP outperforms GNNs. However, it does not discuss how we can modify GNN based on the lesson we learned from AMP-BP. I think the work can be even better if a discussion and preliminary attempt to improve GNNs are included.

---

> ### Author Response · Authors · 2024-01-16
>
> We thank the reviewer for their comments. We want to address the mentioned weaknesses.
>
> Optimality not proven: please see our global answer.
>
> How to design better GNN: this is an interesting point we are working on. We can mention two references ("Line graph neural network" arxiv:1705.08415 and "AMP-inspired deep networks for sparse linear inverse problems" arxiv:1612.01183) that unroll BP and AMP to build new neural networks. We want to combine both. In the conclusion of our revised manuscript, we added the mention of these works and a brief summary of the expected new features of an improved GNN.

---

### Review · Reviewer_S61R · 2024-01-04

**Summary Of Contributions:**

This manuscript proposes an algorithm for classifying nodes in graphs generated under the contextual stochastic block model (CSBM). A conjecture is put forth that the algorithm's predictions approach the ground truth with an error that vanishes in the limit of infinite nodes N and infinite node features P with constant ratio N/P. This conjecture is supported by an argument and experiments.

Finally, the code for this algorithm is provided in supplementary materials, so that it can be used as a benchmark with which to compare popular graph neural network (GNN) graph node classifiers.

**Audience:**

Yes

**Claims And Evidence:**

Yes

**Requested Changes:**

- Equation 2, which defines the generative model, seems to have errors? The writing and notation was difficult to parse.

**Strengths And Weaknesses:**

Strengths:
- The authors make a compelling case for comparing GNN approaches with "simpler" methods. I agree with them that methods such as this could help inspire better GNN algorithms.
- The paper is well-organized and clearly written.
- An implementation of the algorithm is provided.


Weaknesses:
- It is conjectured that the proposed algorithm is optimal, but no proof is provided.
- The conjecture pertains to a particular subsubset of graphs generated by the CSBM.
- The proposed algorithm could return solutions that are far from the optimal. The authors do not discuss these cases and what practitioners should look out for.

---

> ### Author Response · Authors · 2024-01-16
>
> We thank the reviewer for their comments. We would like to address the mentioned weaknesses.
>
> Equation 2: this equation is correct, we clarified the definition of B. Can the referee be more specific about what they found problematic with the equation?
>
> Optimality not proven: please see our global answer.
>
> Indeed this conjecture only applied to graphs generated by the model. We think this still makes the result interesting as data generated by the models are a commonly used benchmark, as we discuss in the introduction.
>
> In terms of "what practitioners should look out for". The way we see the point of our article is not to solve new practical instances but to provide a baseline holding for data generated by the models that practitioners should try to match with new GNN architectures. The algorithm returns solutions that are far from optimal with the probability that decreases exponentially in the system size. In practice, for the sizes of graphs in our simulations, this does not happen. These are, thus, rather reliable benchmarks.

---

> > ### Comment · Reviewer_S61R · 2024-02-13
> > **Still some issues.**
> >
> > The authors have made significant improvements to the article, but I think there are still some obvious problems in the definition of the CSBM model. I had to look up the original definition in Deshpande+ 2018 in order to understand the model. In particular:
> > - In the description of Equation 1, there is an A with subscript mu, nu. I believe this is a typo that should read A_{i,j} to match Equation 1.
> > - The parameter mu appears in Equation 2 without the text explaining what it is.
> > - The description of Equation 2 is also confusing because the authors have swapped the roles of u and nu from the original paper. - It would help clarify the model to state that there are only two centroids in the Gaussian Mixture Model, as done in the original paper.

---

> > > ### Author Response · Authors · 2024-02-19
> > >
> > > We thank the reviewer for these comments. We updated the manuscript with the suggested corrections. Indeed A_{mu, nu} is a typo; we defined the parameter mu and we precised that there are two centroids at ±v in the Gaussian mixture.

---

> > > > ### Comment · Reviewer_S61R · 2024-02-19
> > > > **Thanks**
> > > >
> > > > Thanks! I would add that mu "is a parameter of the model that controls the signal strength". But I am satisfied that description is clear.

---

### Review · Reviewer_SS76 · 2024-01-08

**Summary Of Contributions:**

The authors derive a new belief propagation (BP) algorithm for the semi-supervised node classification problem in attributed networks. This algorithm is conjectured to be optimal in the contextual stochastic block model, a random graph model with community structure and nodes covariates often used to generate synthetic data and benchmark Graph Neural Network (GNN) algorithms. The authors empirically show that there is a gap of performance between the BP algorithm and SOTA GNNs.

**Audience:**

Yes

**Broader Impact Concerns:**

I have no concern about the broader impact.

**Claims And Evidence:**

No

**Requested Changes:**

Requested changes:
- The authors should review the first section (maybe adding a separate section for the literature review) to include references on optimal methods for semi-supervised and unsupervised node classification in the CSBM and closely related models (e.g., feature-adjusted stochastic block model). Some references are included in Sections 2 and 3 but this reduces the overall clarity.
- The authors should correct the introduction of formal notations and clarify the description of the model in Section 2.2. For instance the latent variable v and the group memberships u are not previously defined. Equations 9 and 10 are also unclear: p_i(t) is not defined and I do not understand why (10) gives the MMSE estimator.
- The authors should reformat Section 3: the notations in the diagram at the start of the section are not coherent with the following equations (14), (15), (16). The algorithm on p.5 should be better isolated from the rest of the text and maybe written in single column to make it clearer. Also it is not formal to write “the $\epsilon$s”, etc.
- I think the contributions should be strengthened, e.g., by proving the optimality, a more general version of the algorithm, or by adding experimental results. For instance it would be interesting to compare the BP algorithm in settings with noisy labels or mis-specified settings where it can underperform.
- Some clarifications are needed: What does “approximated asymptotically exactly” on p.8 mean? Why in Figures 4 and 5 the number of layers and neighborhood sizes in the GNNs are not validated hyperparameters?

**Strengths And Weaknesses:**

Strengths:
- The problem of providing an optimal algorithm for comparing the performance of GNNs in setups where they are often applied, i.e., semi-supervised classification and with nodes covariates, is of interest - previously, optimal algorithms often did not include one of the two.
- The paper is generally well-written and easy to follow.
- The novel algorithm has been implemented by the authors and made available (though I cannot see since it is removed by the double blinded review) so that users of GNNs can easily compare to it in their empirical performance study.
- The paper contains an empirical study of the performance of the novel BP algorithm against recent GNNs.

Weaknesses:
- The contribution is slightly incremental, since to my understanding the BP algorithm for the semi-supervised setting is very close to the one for unsupervised classification proposed by Deshpande et al. (2018). Otherwise maybe the differences should be more clearly highlighted.
- The considered setup lacks some justification. Why is the semi-supervised/sparse/high-dimensional setting so important? It is also not easy to follow in the introduction in which (sub)cases there already exist some optimal or supposedly optimal algorithms. The literature review is missing some important references beyond BP algorithms.
- The concept of optimality which qualifies the algorithm is not formally defined in the paper. Besides, except for intuitions on why the algorithm should be optimal in this context, there is no proof or sketch of proof.
- The algorithm is only derived and tested for the case of k=2 balanced communities although the authors underline that an extension to the case of k>2 would be straightforward. An implementation of the general case would add much more value to me.
- There are quite a few imprecisions and things missing in the definitions of the model and notations.

---

> ### Author Response · Authors · 2024-01-16
>
> We thank the reviewer for their comments. We would like to address the mentioned weaknesses and requested changes, in a thematic order.
>
> For the clarity and formalism:
> - we clarified the definition of the model in section 2.1, the variables u and v in section 2.2 and the definition of the marginal p_i(t). (10) is the MMSE estimator because the MSE is $\int dv \sum_u P(u,v|A,B,\Xi) \sum_\beta (\hat v_\beta-v_\beta)^2$ and its minimum is reached for $\hat v_\beta$ equal to the mean of $v_\beta$ over the posterior. We added this precision in the revised manuscript;
> - in the diagram of section 3 we added the indices to the variables to match eqs. 14-16, which we rearranged. We better isolated the algorithm by formatting it in a separate figure and defining the epsilons;
> - we formalized the convergence of AMP–BP to the optimal estimator in section 2.2. By “it can be approximated asymptotically exactly” we mean that the approximation of phi eq. 18 converges in probability to the true phi at large N. We precised this statement.
> - for figures 4 and 5 we did not lead a more extensive search on the number of layers and neighborhood sizes in the GNNs for several reasons. For larger K and l the training becomes harder and does not converge well. For baseline GNNs on figure 4 it appears that the overlaps for K=3 and K=4 are quite close: it seems to reach a limit and for higher K one can expect the features to be over-smoothed and the performance to be worse. For clipGNN figure 5 we added a supplementary curve for smaller N to highlight the limit. This GNN was designed to work on local neighborhoods and one should not take l too large.
>
> As to enriching the literature review:
> - we added a section on related works in the introduction. For optimal algorithms in feature-adjusted SBM would the referee have any particular reference in mind?
> - we are aware of Deshpande et al. (2018); we mentioned it in this review and discuss it in detail in the paragraph "Related work on message passing algorithms in CSBM" of part 3;
> - we added a discussion on the particular semi-supervised, sparse and high-dimensional setting we chose.
>
> On the contribution to be strengthened:
> - we added a new section in appendix where we state the algorithm for r>2 unbalanced communities;
> - we already provide a version of the algorithm for dealing with mis-specified settings in appendix B, eqs. 79-81;
> - dealing with noisy labels is straightforward in our Bayesian setting. One has to replace the prior P_{U,i} by the actual distribution of the labels. We added a comment on this.

---

### Author Response · Authors · 2024-01-16

We updated the manuscript and the appendix following some of the comments, questions and suggestions; the changes are highlighted.

We answer to a point raised by the three reviewers, that the optimality of AMP–BP is not proven. We would like to emphasize that the optimality of BP for the simpler sparse SBM is still a conjecture (for over a decade already), and solving it would be a major result on its own.

The optimality of AMP for the Gaussian mixture model follows from existing work, as cited in Section 3. In Section 3 we also provide an argument for why joining the two models together and proving the optimality of the joint AMP-BP would then follow rather readily. The bottleneck really is the proof of optimality of BP for the SBM which is a very ambitious endeavour that we did not attempt to solve.

---

### Decision · Action_Editor_LrcB · 2024-02-21

**Recommendation:** Accept as is

**Comment:**

The paper was reviewed by three expert reviewers. All reviewers recommended acceptance of the paper and I agree with them. The reviewers initially complained about the clarity of the paper, and the authors made efforts to improve the presentation in the revision. All three reviewers also raised concerns regarding the use of the term "optimal" in the title and throughout the paper. The authors made clear in their response that the optimality of BP for the simpler sparse SBM has not been proven yet and is still a conjecture. I am thus recommending that the authors modify the title of the paper and replace the word "optimal", which seems somewhat misleading, with other terms/phrases that indicate that optimality actually remains a conjecture. For example, the authors could use the terms "towards optimal inference", but I leave this decision to the authors.

**Audience:**

The topic and findings of the paper are of interest to some individuals in TMLR's audience, mainly individuals working in the field of graph representation learning. The authors show that there is a considerable gap between the performance of the proposed method and that of GNNs on the CSBM dataset, and this gap might inspire a range of follow-up works.

**Claims And Evidence:**

In this paper, the authors propose AMP–BP, a belief-propagation-based algorithm for the semi-supervised contextual stochastic block model (CSBM). The authors conjecture that the proposed algorithm is optimal in the considered setting. They also claim that in this setting, there can be a considerable performance gap between the proposed algorithm and graph neural network (GNN) models. Their claim is supported by convincing and clear evidence.

---

> ### Author Response · Authors · 2024-03-05
>
> We thank the reviewers and the editor for their work. In the camera-ready revision we added a reference to an article proving the optimality of BP on sparse semi-supervised binary SBM we were not aware of before and that supports our conjecture. We kept the previous title.